# A Simplified Framework for Modelling Viscoelastic Fluids in Discrete Multiphysics

Carlos Duque-Daza [1,2,*] and Alessio Alexiadis [1,*]

1   School of Chemical Engineering, University of Birmingham, Birmingham B152TT, UK
2   GNUM Research Group, Universidad Nacional de Colombia, Bogotá 111321, Colombia
*   Correspondence: caduqued@unal.edu.co (C.D.-D.); A.Alexiadis@bham.ac.uk (A.A.)

**Abstract:** A simplified modelling technique for modelling viscoelastic fluids is proposed from the perspective of Discrete Multiphysics. This approach, based on the concept of linear additive composition of energy potentials, aims to integrate Smooth Particle Hydrodynamics (SPH) with an equivalent elastic potential tailored for fluid flow simulations. The model was implemented using a particle-based software, explored and thoroughly validated with results from numerical experiments on three different flow conditions. The model was able to successfully capture a large extent of viscoelastic responses to external forcing, ranging from pure viscous flows to creep-dominated Bingham type of behaviour. It is concluded that, thanks to the modularity and tunable characteristics of the parameters involved, the proposed modelling approach can be a powerful simulation tool for modelling or mimicking the behaviour of viscoelastic substances.

**Keywords:** Discrete Multiphysics; viscoelasticity; viscoleastic fluids; Smooth Particle Hydrodynamics; coarse-grained molecular dynamics; viscoelasticity modelling





## 1. Introduction

Viscoelastic materials combine mechanical properties typical of solids (i.e., elasticity) and fluids (i.e., viscosity). For instance, many polymer substances exhibit both viscous and elastic properties during deformation of the material. For these substances, instead of a well defined elastic or viscous response to deformation, a time-dependent or shear-dependent strain can be observed. Although polymers can be taken as a clear example of viscoelastic substances, there are many other materials that also exhibit a large range of time-dependent stress–strain behaviour [1–3]. Viscoelasticity, the field devoted to study viscoelastic behaviour, is an important tool to understand a large number of physical processes, including molecular mobility in polymers [4], and analysis of dynamics of defects in crystalline interfaces in solids [5]. It is also instrumental in the process of designing new materials and devices employed for a variety of purposes, including vibration abatement, control of mechanical shocks and vibrations, and noise reduction [6]. Viscoelastic flows are important for a great number of industrial applications, as well as present in many common daily situations. For instance, the behaviour ranges from toothpaste flowing by extrusion [7], to metals or molten polymers casting [8], and includes many food industry production processes and analysis [9,10]. The viscoelastic behaviour has been recognised as a dominant and extremely important characteristic of many practical flows, and therefore a key phenomenon that needs to be better understood.

Viscoelasticity is one of the main subjects of rheology, a field of study concerned with the description of the flow of matter and the mechanical properties of substances under various deformation conditions. In rheology experimental methods have usually been the main source of data for the analysis of the flow-type response of matter [11]. Nonetheless, to understand and effectively use available experimental rheological information, it is essential to have a consistent mechanistic framework. Moreover, interpretation of viscoelastic behaviour in terms of clear theoretical concepts should produce guidelines to

make a clean sense of observations, to relate behaviour to composition and structure, to predict or estimate physical properties, and to facilitate control of practical applications of viscoelastic substances. In spite of the large number of current practical uses of viscoelastic materials, there are not simple expressions or models that could be considered as a generalised mechanistic framework that fit all experimental data. Although the body of studies on modelling viscoelastic fluids is large, it is still relatively modest compared to the extension of research dedicated to other "more conventional" fluid flow cases.

Modelling is a valid alternative to explore complex viscoelastic materials, to complement any experimental approach and a field to be further explored [12]. For instance, a new spectral modelling approach has been recently proposed where constitutive equations are formulated in terms of spectral invariants (see [13]), and which has been employed to formulate three-dimensional constitutive modelling framework employing a viscoelastic matrix to model residually stressed viscoelastic solids using the finite element method [14]. However, modelling viscoelastic fluids is not an immediate or simple task. The complex nature of the viscoelastic flows has somehow precluded attempts to obtain a general and accurate model. Some of this complexity can be perceived in the number of theoretical models that have been brought about to capture the extended range of possible cases [15–18], which seems to confirm that obtaining a simple universal model is far from easy. Mackay and Phillips [19] highlight that although the experimental contributions to the characterization of polymeric materials in recent decades have been abundant, there seems to be a lack of similar momentum in the development of modelling techniques for those type of fluids. Some examples on viscoelastic behaviour modelling include the use of particle-based viscoelastic fluids modelling [20], modelling based on the finite element method coupled with the generalised bracket method [19], continuum models formulated in terms of tensor diffusion and the Tensor Stokes problem [21], and the use of a formulation based on the Gibbs-potential aiming to obtain thermodynamically consistent modelling of viscoelastic fluids [22]. Besides these rigorous but computationally expensive approaches, viscoelasticity is also of interest for computer graphics [23–25]. In this case, the focus is not so much on the accuracy of the physical model, but rather on obtaining a fast simulation that preserves the overall visual effect of the viscoelastic material.

In the present work, we propose an alternative modelling technique for viscoelastic fluids within the Discrete Multiphysics (DMP) framework (see [26]) that is somehow in between the rigorous approach and computer graphics. DMP is an alternative hybrid approach for modelling multiphysics phenomena in which particle-based methods, e.g., Smooth Particle Hydrodynamics (SPH), are combined with coarse-grained molecular dynamics to capture a wide range of material's behaviour, and which has been particularly successful in modelling multiphysics and multi-phase problems with large interfacial deformations (see [26–30]). In the present work we do not directly modify the equations of motion to account for viscoelasticity, but we build a particle model where force fields typically used for modelling elastic and viscous materials are coupled together. This approach is easy to implement and more accurate than the one used in computer graphics since the viscoelastic properties of the fluid can be verified in detail. However, it loses the ability to directly derive the viscoelastic property of the model from first principles. Thus, it is required to establish these properties ex-post by performing a series of numerical experiments. In [31], a similar approach was used for viscoelastic solids; here, it is extended to viscoelastic fluids.

The paper is organized as follows. In Section 2, we discuss several theoretical concepts related to viscoelasticity. After that, we introduce Smooth particle Hydrodynamics (SPH, Section 3) used for modelling the viscous behaviour and the potentials (Section 4) used for modelling the elastic behaviour. Subsequently in Section 5, three benchmark cases are chosen to validate the results and to explain how the modelling approach works in practical settings.

## 2. Viscoelastic Behaviour and Standard Models

A compelling feature associated with the deformation of a viscoelastic substance is its simultaneous display of "fluid-like" and "solid-like" characteristics [32]. Arguably, one of the most important features to examine in a fluid-like substance, regardless of its Newtonian or non-Newtonian nature, is the response to shear forcing. In fact, this is the property that has traditionally been taught as the ultimate differentiating element between fluids and solids: if the substance is able to stand shear forcing, without large or permanent deformation, it is generally classified as solid, whereas if the substance is unable to stand shear forcing, thus permanently being deformed, then it is considered a fluid [33]. However, this description is not complete, as there exist a range of materials and substances that clearly show a blended behaviour between pure elastic or pure viscous. This dual response, clearly distinguishable in many aspects of the behaviour of certain substances to external forcing (i.e., shearing), has motivated many of the attempts to describe the viscoelastic behaviour in terms of basic mechanistic models, through which a simple description was always pursued. For instance, in the traditional view of elasticity, the stress found in a substance undergoing deformation is directly proportional to the strain, so the traditional applicable model is the Hooke's law that, in tension, reads as

$$\tau_{ij} = -G\frac{\partial X_j}{\partial x_i} = -G\gamma_{ij} \tag{1}$$

where $G$ is known as Young's modulus, $X_j$ is the shear displacement in any given $j$-th direction of two elements separated by an infinitesimal gap in the $x_i$ direction, and $\gamma_{ij}$ is the shear strain. If a deformed substance is able to recover its original shape once the stress is withdrawn, then Equation (1) is an appropriate model for characterizing the elastic mechanical response of the system. Most of the elastic substances might exhibit also a threshold stress beyond which the substance will "flow", and a complete recovery of the shape is not longer possible, a condition known as creep. On the other hand, a Newtonian fluid, the most representative example of a pure viscous substance, will show a shear stress proportional to the shearing rate, and for which a standard simple model can be written as the Newton law of viscosity,

$$\tau_{ij} = \mu\frac{\partial \dot{X}_j}{\partial x_i} = \mu\dot{\gamma}_{ij} \tag{2}$$

where $\mu$ is the dynamic viscosity, a proportionality constant in Equation (2), $\dot{X}_j$ is the velocity displacement in any given $j$-th direction, and $\dot{\gamma}_{ij}$ is the shear strain rate. The fact that the stress in a pure elastic substance is directly defined by the strain by virtue of Equation (1), whereas in a pure viscous substance is proportional to the strain rate (Equation (2)), brings about a phase synchronisation/de-synchronisation between strain and stress when a substance is subject to an oscillating strain. As a simple example, if a material is subject to a periodic shearing strain, for instance a sinusoidal function $\gamma = \gamma_0 \sin(\omega t)$, with angular frequency $\omega$ and amplitude $\gamma_0$, then the elastic substance will present a stress in phase with the strain. A viscous substance, subject to the same periodic strain, will instead exhibit a 90°-out-of-phase in time stress signal with respect to the same oscillating strain (see Figure 1).

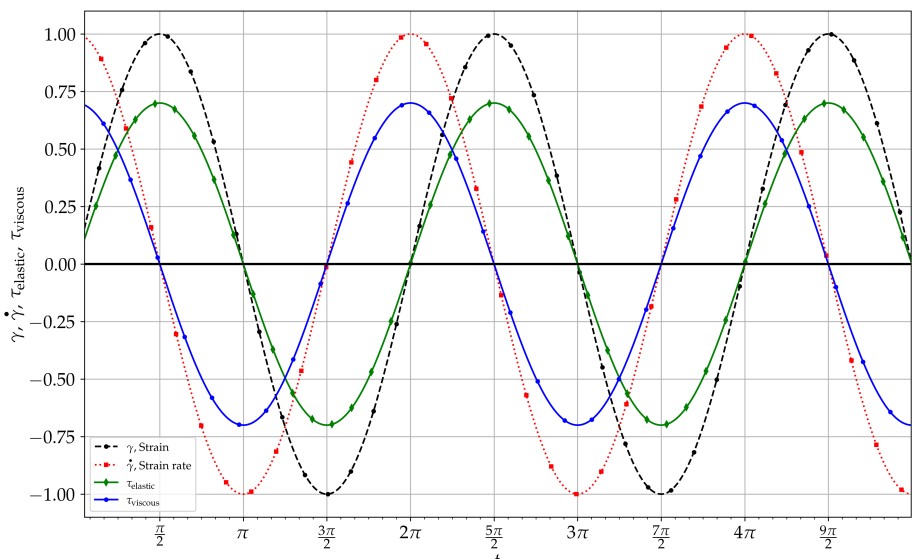

**Figure 1.** Strain rate and stress of ideal elastic and viscous substances under an oscillating strain $\gamma = \gamma_0 \sin(\omega t)$.

The outlined mechanical response can be somehow captured by using two simple mechanical analogues, i.e., by using a "spring" model to represent an elastic behaviour, and a raw "dashpot" model for representing a viscous response. On the basis of these two simple mechanical models, and aiming to get a quantitative description of the viscoelastic behaviour, the rationale behind many of the models that have been formulated is that by a simple coupling process of the crude mechanical analogues, for instance in an additive way, the more complex response of viscoelastic substances could be attained [34–36]; some of the most known models for modelling viscoelastic substances are Maxwell, Kelvin, Kelvin–Voigt, and Burgers models. Schematic representations of three of these models are presented in Figure 2. Examples of use and applications of these models can be found in [37–39] for the Maxwell model, in [40] for the Kelvin–Voigt and Maxwell fractional models, and in [41] for the Kelvin–Voigt and Burgers models, just to cite a few.

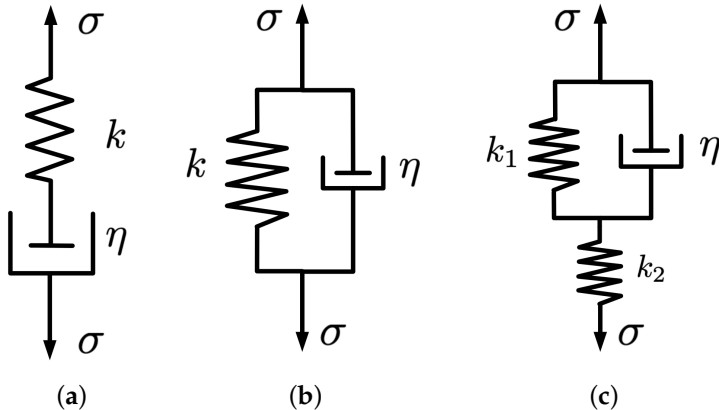

**Figure 2.** Schematic representation of some standard models based on mechanical analogues. (**a**) Maxwell model. (**b**) Kelvin model. (**c**) Kelvin–Voigt model.

In the Maxwell model the spring and dashpot are arranged in a serial configuration (see Figure 2a), while in the Kelvin model the spring and dashpot are arranged in parallel (see Figure 2b). The simple arrangements of these two models, for example, produce constitutive equations than can be written as [35]

$$\sigma(t) = k\epsilon(t) + \eta\dot{\epsilon} \tag{3}$$

for the Maxwell model, and

$$\sigma(t) + \frac{\eta}{k}\frac{d\sigma(t)}{dt} = \eta\dot{\epsilon} \tag{4}$$

for the Kelvin model, where $\sigma$ is stress, $\epsilon$ is the strain, $\dot{\epsilon}$ is the strain rate, $k$ is the characteristic Hooke elastic spring constant, and $\eta$ is the equivalent Newtonian viscous dashpot constant. In a general oscillating strain condition, for instance if we prescribe $\epsilon = \epsilon_0 \sin(\omega t)$, and excluding transient periods, it is possible to see that both models, Maxwell and Kelvin, predict similar stress responses in time, although with variations in amplitude and out-of-phase angle with respect to the strain signal. As an illustrative case, if the constant of the viscous component is set as $\eta = k/\omega$, this configuration produces similar stress signals for both models that, in general, are out-of-phase with respect to the driving oscillating strain, although they are in phase to each other, as shown in Figure 3. On the other hand, if the ratio between $\eta$ and $k$ is changed, the same oscillating strain will produce a variety of responses for the Maxwell and Kelvin models. For instance, if the dashpot constant is set as $\eta = 0.5k/\omega$, it is possible to observe phase difference between the stress response predicted by the Maxwell model and the stress predicted by the Kelvin model, as presented in Figure 4, although both stress responses still exhibit a phase difference with the strain signal. This particular configuration brings about an almost elastic response in the Maxwell model, but a more dissipative behaviour in the Kelvin model. This situation is inverted if the dashpot constant is defined as $\eta = 2.0k/\omega$, as shown in Figure 5.

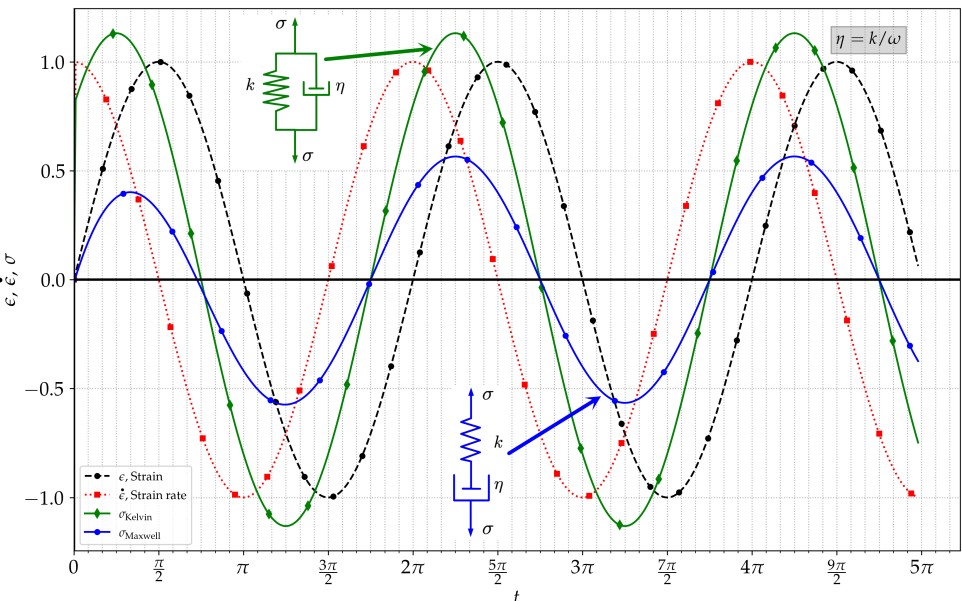

**Figure 3.** Strain, strain rate, and stress evolution in time for ideal Maxwell and Kelvin models forced with a synthetic strain $\epsilon = \epsilon_0 \sin(\omega t)$. Curves for $\epsilon = 1.0$, $\omega = 1.0$, $k = 0.8$, $\eta = k/\omega$.

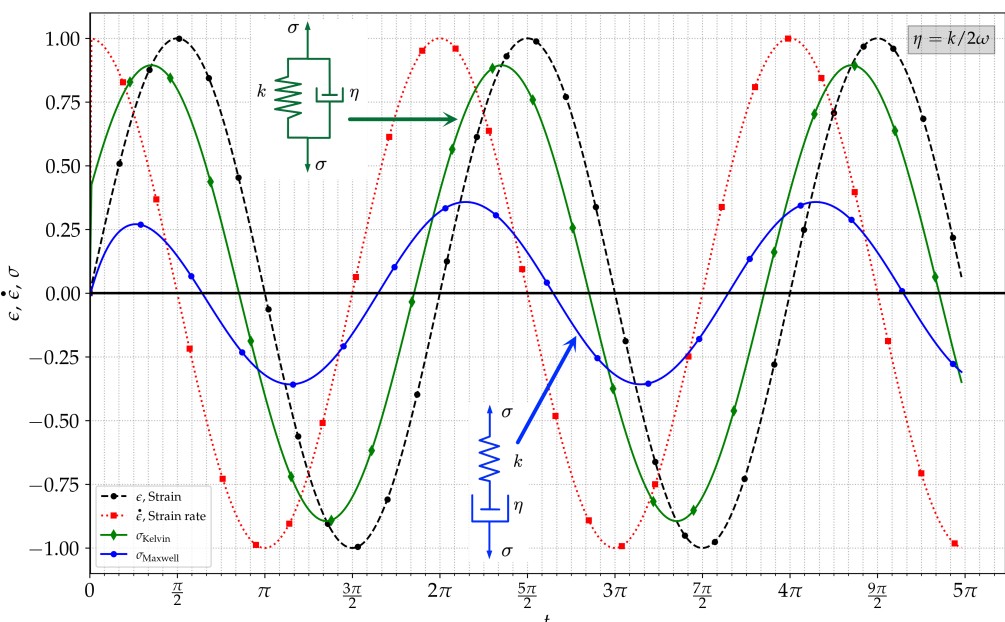

**Figure 4.** Strain, strain rate, and stress evolution in time for ideal Maxwell and Kelvin models forced with a synthetic strain $\epsilon = \epsilon_0 \sin(\omega t)$. Curves for $\epsilon = 1.0$, $\omega = 1.0$, $k = 0.8$, $\eta = 0.5 k/\omega$.

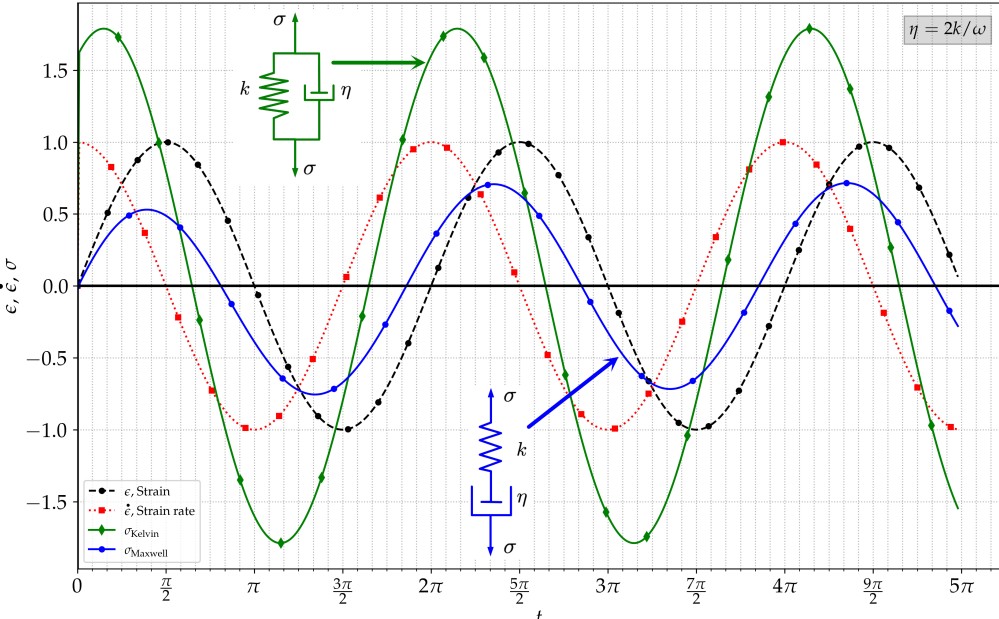

**Figure 5.** Strain, strain rate, and stress evolution in time for ideal Maxwell and Kelvin models forced with a synthetic strain $\epsilon = \epsilon_0 \sin(\omega t)$. Curves for $\epsilon = 1.0$, $\omega = 1.0$, $k = 0.8$, $\eta = 2.0 k/\omega$.

It is important to highlight that the aforementioned models are valid for linear or quasi-linear viscoelastic behaviour, i.e., assuming that the substance is undergoing small deformations. If linear viscoelasticity is adopted, the stress function $\sigma(t)$ and the strain function $\epsilon(t)$ can de considered as linear proportional models, so if the strain function $\epsilon(t)$ is amplified by a constant factor, the outcoming stress would be scaled by the same factor, and if a substance is subject to a linear combination of two (or more) arbitrary strain signals, the stress can be obtained as the linear combination of the two (or more) individual stress responses [35]. The out-of-phase stress response, captured by the linear viscoelastic models, can be gauged through the so-called loss angle ($\delta$), the phase angle between stress and strain during sinusoidal deformation in time, as the example cases presented in Figures 3–5.

The use of the loss angle or its tangent ($\tan \delta$), is extremely useful to produce a measure of damping or internal friction when linear viscous nature is assumed. The loss angle (or $\tan \delta$), depends mostly on the frequency of the external excitation. Although the discussion for the viscoelastic models has been based on a general normal stress $\sigma$ and longitudinal strain $\epsilon$, it is clear from Equations (1) and (2) that the whole discussion can easily and immediately be extended to a case of a pure oscillating shear applied to a substance. It is also customary to express the oscillatory behaviour of linear viscoelastic substances in complex notation, so a pure oscillating shear condition can be obtained if a shear strain $\gamma(t) = \gamma_0 e^{\mathbf{i}\omega t}$ is imposed, which will produce a shear stress response equal to $\tau(t) = G e^{\mathbf{i}\omega t}$. The resulting coefficient $G$ will be frequency-dependent and in general a complex number that can be expanded as,

$$G(\omega) = G'(\omega) + \mathbf{i} G''(\omega) \tag{5}$$

and from which it is possible to discriminate between the real component $G'(\omega)$, associated to the elastic part of the response, and the complex component $G''(\omega)$, usually considered as a quantification of the viscous part of the response. The first component is called the Storage Modulus, while the complex part is known as the Loss Modulus.

## 3. SPH Formulation

In this study, we combine different particle methods together. SPH, discussed in this section, is used for viscous behaviour and other specific potentials (see next Section) for the elastic behaviour. This approach of combining different particle methods is called Discrete Multiphysics (e.g., see [26–28]) and it is here extended to account for viscoelastic fluids. The numerical method used in the present work to model the fluid is Smooth Particle Hydrodynamics, or simply SPH, an approximate method to obtain numerical solutions to the equations of fluid dynamics. This is done by replacing the continuum of fluid by a discrete set of particles. It was originally devised for the simulation of stars [42,43], and was later found applicable to molecular dynamics (MD) due to the inherent similarity between SPH and MD. To reproduce the equations of fluid dynamics or continuum mechanics, the statistical concept of kernel interpolation is used to 'smooth' out discrete fields of a quantity of interest (such as density, pressure and velocity). Moreover, in SPH each particle is represented by a kernel function (more generally a window function) $W(\vec{r} - \vec{r}', \vec{h})$. The local average of the desired property $f$, in a domain of interest $\Omega$, is the convolution of the quantity $f$ with the chosen smoothing function $W(\vec{r} - \vec{r}', \vec{h})$,

$$\langle f(\vec{r}) \rangle = \int_{\Omega} f(\vec{r'}) W(\vec{r} - \vec{r'}, \vec{h}) \, d^3\vec{r'} \tag{6}$$

The kernel functions considered in SPH are generally radially symmetric (spherical) functions centred at $\vec{r'}$ and must decrease monotonically in the outward radial direction. These kernel functions must be normalised to 1 so that constants are interpolated exactly,

$$\int_{\Omega} W(\vec{r} - \vec{r'}, \vec{h}) \, d^3\vec{r'} \to 1 \quad \text{as} \quad h \to 0. \tag{7}$$

and they should tend to a Dirac delta function $\delta(\vec{r} - \vec{r'})$ as the smoothing length tends to zero, $h \to 0$, in order to recover the original function $f$ in the limit.

$$\langle f(\vec{r}) \rangle \to f(\vec{r}) \quad \text{as } h \to 0 \tag{8}$$

The chosen smoothing length $h$ is often a characteristic length of the target domain $\Omega$. In general, it is advisable for this smoothing length to be a constant value for the prescribed problem so that its spatial and temporal derivatives are identically zero, although variable smoothing lengths in time and space can provide simulation resolutions that adapt to local conditions. It is important to note that the shape of the constructed continuous field will be fully determined by the choice of the kernel function $W(\vec{r} - \vec{r'}, \vec{h})$, the smoothing length

$h$, and the position of each particle in the domain of interest i.e., the particle distribution, as schematically represented in Figure 6. In SPH the interest is focused on the cases when neighbouring particles are each encircled by the radius of the smoothing length $h$, as any other particles outside this radius would not be considered by the compact support of the kernel function, and so would generate a discontinuous field. This can be easily enforced by setting the smoothing length to be an appropriate characteristic length.

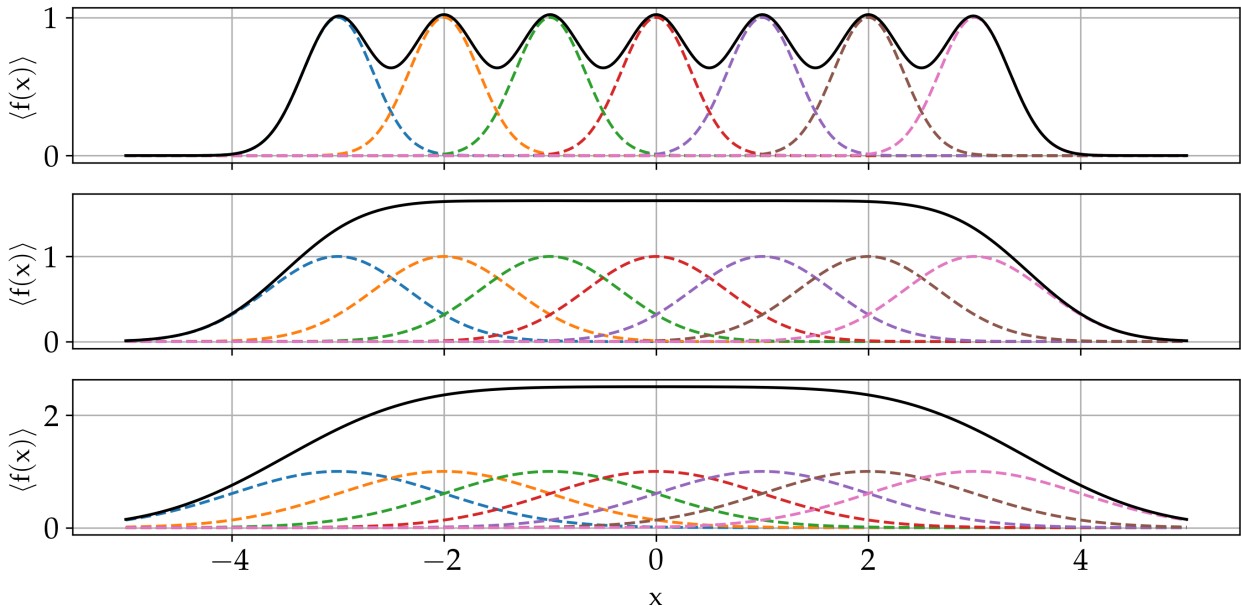

**Figure 6.** Conceptual example of the effect that the smoothing length has on $\langle f(x) \rangle$ for multiple point-particles, separated one length unit, and with Gaussian function as kernel. The smoothing lengths employed in the examples are: $h = 1$, upper plot; $h = 2$, middle plot; and $h = 3$, bottom plot.

For computational interest, we can discretise Equation (6) as follows,

$$\langle f(\vec{r}) \rangle = \sum_{i=1}^{N} f(\vec{r}_i) W(\vec{r} - \vec{r}_i, \vec{h}) V_i \tag{9}$$

where $V_i$ is the volume of the particle of interest and $N$ is the number of particles within the domain. An equivalent way of this discretisation, common in fluid dynamics applications, can be expressed as

$$\langle f(\vec{r}) \rangle = \sum_{i=1}^{N} \frac{m_i}{\rho_i} f(\vec{r}_i) W(\vec{r} - \vec{r}_i, \vec{h}) \tag{10}$$

in which $\rho_i$ is the mass density of the i-th particle (considered a local density) and $m_i$ its mass. By using index notation this can be simplified further as,

$$\langle f_i \rangle = \sum_{j=1}^{N} m_j \frac{f_j}{\rho_j} W_{ij} \tag{11}$$

with $W_{ij} = W(\vec{r}_i - \vec{r}_j)$ and $f_i = f(\vec{r}_i)$. For fluid dynamics applications it is also highly relevant to define appropriate discrete forms of the gradients present in the governing equations of motion. It is noteworthy that SPH is rather convenient in that neither $m_i$ or $f_i$ are affected by the $\nabla$ operator since they are particle properties. The kernel function is

usually a polynomial so we can generate an expression for the gradient by simply taking the gradient of $W_{ij}$,

$$\nabla f_i = \sum_{j=1}^{N} m_j \frac{f_j}{\rho_j} \nabla W_{ij} \tag{12}$$

As a simple example, we demonstrate the SPH approximation of the continuity equation,

$$\frac{d\rho}{dt} + \rho \nabla \cdot \vec{v} = 0 \tag{13}$$

which, by using the identity $\nabla(\vec{v}\rho) = \rho\nabla\vec{v} + \vec{v}\nabla\rho$, can be rewritten as,

$$\frac{d\rho}{dt} = \nabla(\rho\vec{v}) - \vec{v}\nabla\rho \tag{14}$$

and so using the gradient approximation we have:

$$\frac{d\rho_i}{dt} = \sum_{j=1}^{N} m_j \vec{v}_j \nabla_j W_{ij} - \vec{v}_i \sum_{j=1}^{N} m_j \nabla_j W_{ij} = -\sum_{j=1}^{N} m_j \vec{v}_{ij} \nabla_j W_{ij} \tag{15}$$

Discretised SPH forms of the other relevant conservation equations are widely available in the literature (see [44–46]).

The numerical implementation of our model, including the SPH-viscous component, was performed using LAMMPS [47,48]. In LAMMPS, the SPH method is achieved through the use of pair styles defining the interaction between neighbouring particles following the SPH formulation. For liquids, the two most common SPH models are the compressible model, proposed by [44] apt for high speed flows, and the model proposed by [49], usually better suited for low Reynolds number incompressible flows. One of the main differences between the two models lies in the role of the viscosity within the SPH simulation, which appears naturally in the equations of conservation of momentum in flow dynamics as part of the relationship between stress and strain rate. In SPH, the viscosity is incorporated into the numerical formulation in a number of different ways [44,46]. Nevertheless, some physical phenomena have been more challenging for SPH, requiring the use of numerical strategies aiming to minimize problems like numerical spurious oscillations (generally around shock regions) or unphysical particles penetration, and usually through the introduction of energy dissipation strategies that exploit the viscosity as a dissipative tool [50–52]. In the model proposed in [49], the dynamic viscosity is one of the parameters to be provided, together with the smoothing length $h$, the speed of sound $c$, and the density $\rho$. Alternatively, a common artificial viscosity employed in SPH, introduced in [44] originally as an extension of the von Neumann–Richter artificial viscosity, is defined as

$$\Pi_{ij} = -\alpha h \frac{c_i + c_j}{\rho_i + \rho_j} \frac{\mathbf{v}_i \cdot \mathbf{r}_{ij}}{r_{ij}^2 + \epsilon h^2} \tag{16}$$

where $c_i$ and $c_j$ represent the speed of sound of particles $i$ and $j$, $\epsilon$ is a constant to avoid singularities in the definition of the viscous component $\Pi_{ij}$, and $\alpha$ is a dissipation factor that serves as a modulating coefficient of the viscous response from the SPH perspective. A widely accepted equivalence between the real dynamic viscosity $\mu$ and the dissipation factor $\alpha$ is given as,

$$\mu = \frac{\alpha c h \rho}{2(n+2)} \tag{17}$$

where, as usual, $c$ represents a numerical speed of sound, $h$ the smoothing length, $\rho$ the density of the target fluid and $n$ is the number of spatial dimensions involved. It is noteworthy that in our DMP approach point-particles rather than finite-size particles are employed, although particle volume is defined to correlate particles mass and density.



## 4. Proposed Modelling Technique

Viscoelasticity can be directly implemented in SPH (e.g., [53]). However, this implies rewriting the equation of motion to account for a specific model of viscoelasticity. In the case of a particle code like LAMMPS, for instance, this would imply rewriting large sections of the code dedicated to SPH every time we introduce a new model of viscoelasticity. The approach proposed in this study models viscoelasticity by combining different particle potentials, which is a standard procedure in particle codes, rather than rewriting the equations of motion. For the viscous part, the standard SPH approach discussed in the previous section is adopted. The elastic part is discussed in this section.

In an analogue manner to the idea supporting the most widely used basic viscoelastic models discussed previously, we propose an alternative modelling technique within a particle-based framework, where inter-particle potentials tackling viscous and elastic interactions separately are blended in an additive way in order to mimic the dual dissipative and restoring response of viscoelastic substances. For this purpose, the SPH viscous support is blended with a potential that resembles a restorative elastic behaviour, aiming to bring about the characteristic dual response of viscoelastic substances. In our approach, the viscous properties are given to the fluid by the SPH Equations (16) and (17). Furthermore, in principle, we would like to use harmonic springs that are used in the Lattice Spring Model (LSM) to add elasticity to the material (see [54]),

$$U_{\text{LSM}}(r) = \frac{1}{2}k(r - r_0)^2 \tag{18}$$

which would provide a force $F(r) = -\nabla U(r) = -k(r - r_0)$ between two particles connected by the spring with a stiffness $k$, separated a distance $r$, and featuring an equilibrium distance $r_0$. The combination of Equations (16) and (17) and LSM will confer viscoelastic properties to the material. A similar approach to model viscoelastic solids was employed in [31]. However, fluids cannot be modelled in the same way because the springs constrain the particles, which cannot flow as it should occur in fluids. The solution we propose is to employ a blending of two potentials to partially imitate the LSM, but with the advantage of featuring a deactivation or cutoff distance beyond which the composed potential not longer acts, allowing the particles to flow freely. An exclusively attractive potential, first proposed by [55], was chosen to provide the attractive portion of our LSM analogue, defined as,

$$U_{\text{CS}}(r) = \begin{cases} -\epsilon_{\text{CS}} & r < \sigma \\ -\epsilon_{\text{CS}} \cos\left(\dfrac{\pi(r - \sigma)}{2(r_{c,\text{CS}} - \sigma)}\right)^2 & \sigma \leq r < r_{c,\text{CS}} \\ 0 & r \geq r_{c,\text{CS}} \end{cases} \tag{19}$$

which, as schematically illustrated in Figure 7a, is a constant value of $\epsilon_{\text{CS}}$ below an inter-particle distance $\sigma$, that increases proportional to $r$ until it vanishes above a cutoff distance $r_{c,\text{CS}}$, according to Equation (19). This allows us to account for the fact that particles may no longer feel an attractive force when separated far apart; thus fluid particles are not constrained into a lattice structure like in Equation (18), but they are free to flow within the domain.

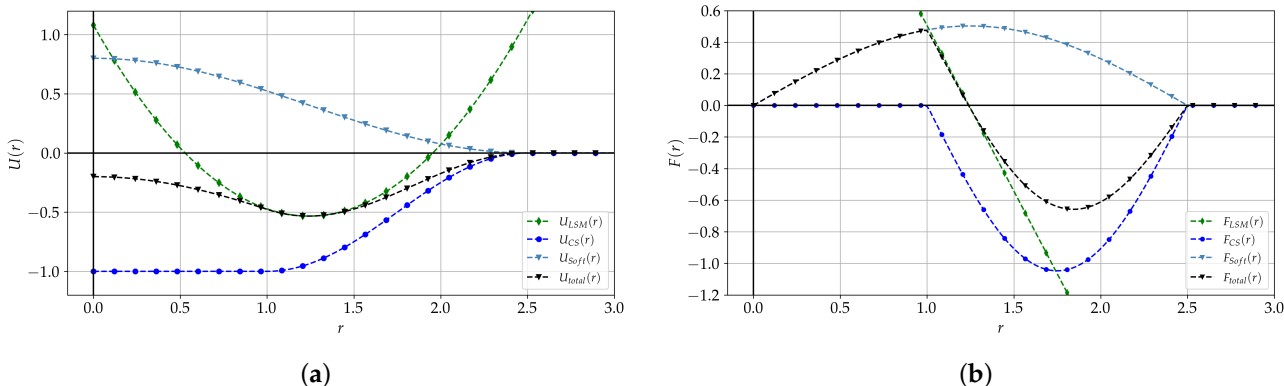

**Figure 7.** Comparison of energy potentials and forces used for modelling elastic behaviour. *LSM*: Lattice-Spring-Model potential; *CS*: Attractive potential (Cosine/squared); *Soft*: Repulsive potential (Soft); *Total*: Combined Soft-Cosine/Squared potential. Curves obtained for $\epsilon_{CS} = 1$, $\epsilon_{Soft} = 0.4\epsilon_{CS}$, $\sigma = 1$, $r_{c,CS} = r_{c,Soft} = 2.5$. (**a**) Energy potentials. Location of the harmonic LSM potential adjusted for illustration purposes. (**b**) Force field obtained with elastic, attractive, repulsive, and total potentials.

To avoid particle overlap, a repulsive portion in our model is provided by another potential given as,

$$U_{\text{Soft}}(r) = \epsilon_{\text{Soft}} \left[ 1 + \cos \left( \frac{\pi r}{r_{c,\text{Soft}}} \right) \right] \qquad r < r_{c,\text{Soft}} \tag{20}$$

where, $\epsilon_{\text{Soft}}$ is the magnitude and $r_{c,\text{Soft}}$ is the cutoff distance for this repulsive component of our potential therefore, once again, limiting the distance at which repulsion exists for particles far apart. In our proposed model we use a total potential by adding these components aiming to mimic the elastic behaviour of the LSM model, but with the advantage that when the distance between the two particles is above the cutoff value, the force is deactivated and the particle is free to flow. An illustration of the LSM, repulsive, attractive, and total potentials is presented in Figure 7a, while a representation of the forces produced by them is shown in Figure 7b, where the repulsive and attractive components of the elastic equivalent forces are schematically depicted.

As the rationale behind our model is to be able to get a coupling between the two pair styles producing stable repulsion/attraction within a close range of each particle, this coupling requires a sensible choice of values given the possible complex interaction that can be brought about by the number of parameters at play. For instance, an important consideration, usually critical in other mesh-based methods only from the numerical stability perspective, is the spatial resolution. In our case, the spatial resolution is somehow linked to the distribution of particles, so the stability and effectiveness of the model is also dependent of the lattice employed for the initial arrangements of such particles, as well as on the characteristic length in the lattice, or lattice scale $\Delta_L$. Specifically, in order to reduce the number of free parameters, the repulsive potential was defined in terms of the characteristics of the attractive potential, so only a single set of values could define entirely the elastic coupled model. It is important to mention that, as our model is strongly dependent of the inter-particle spacing, the data presented along the present paper as reference values should be taken as a guide for setting up working simulations, rather than as a restrictive range of operating conditions. In any case, some ranges and definitions employed in the characterization of the elastic component of our model, determined through several numerical experiments and found to provide numerical stability and consistency with expected behaviour, are presented in Table 1 for reference purposes. In Table 1 the acronyms *BCC* and *FCC* stand for Body-centred cubic and Face-centred cubic unit cells, respectively, which are some of the most common type of regular lattices for molecular dynamics simulations.

**Table 1.** Summary of some ranges and relationships between parameters for the elastic potential.

|  | BCC Lattice | FCC Lattice |
|---|---|---|
| Activation distance-attractive potential | $\sigma = \sqrt{3}\,\Delta_L/2$ | $\sigma = \sqrt{2}\,\Delta_L/2$ |
| Cutoff distance-attractive potential | $0.95\,\Delta_L \leq r_{c,\text{CS}} \leq 2.1\,\Delta_L$ | $0.9\,\Delta_L \leq r_{c,\text{CS}} \leq 1.1\,\Delta_L$ |
| Cutoff distance-repulsive potential | $r_{c,\text{Soft}} \approx 1.05\,r_{c,\text{CS}}$ | $r_{c,\text{Soft}} \approx 0.955\,r_{c,\text{CS}}$ |
| Prefactor-repulsive potential | $0.8\,\epsilon_{\text{CS}} \leq \epsilon_{\text{Soft}} \leq 3.0\,\epsilon_{\text{CS}}$ | $1.25\,\epsilon_{\text{CS}} \leq \epsilon_{\text{Soft}} \leq 4.0\,\epsilon_{\text{CS}}$ |

## 5. Numerical Experiments

The proposed model was characterised and tested using mainly two types of conditions: oscillatory shearing between parallel plates, and constant gradient in a circular pipe flow. Numerical simulations with two types of lattice, namely *Body-centred cubic* or BCC, and *Face-centred cubic* or FCC, were performed for both conditions, although results reported here are mainly based on the numerical experiments using the FCC lattice. Regardless of the lattice employed, the final number of particles in each of our numerical experiments was obtained following a same methodology, i.e., preliminary exploratory simulations were performed first to get time evolution of some representative local measures, like velocity and force ensemble averages, together with an assessment of the level of "mesh independence", and the final selection of the number of particles for each test was decided with basis in those results. Statistical convergence was assumed once the mean and standard deviation of the ensemble averages converged in the preliminary simulations. Furthermore, it is important to mention that although the proposed methodology is general and easy to implement with any SPH and coarse-grained molecular dynamics software, the numerical experiments of the present work were performed using LAMMPS (see [47,48]). The computations described in this paper were completed using the University of Birmingham's BlueBEAR HPC service, and executed in parallel using LAMPPS' MPI capabilities.

### 5.1. Dynamic Response to Oscillating Shear

The most traditional way to differentiate the viscous from the viscoelastic behaviour is through the determination of the storage and loss moduli presented before. These quantities should be obtained from oscillatory shear tests. To this end, a substance contained between two parallel plates, one of them stationary and the other one oscillating at a fixed frequency, was modelled using our viscoelastic model. The domain for this numerical experiment is a box of dimensions $0.30\,\text{m} \times 0.14\,\text{m} \times 0.02\,\text{m}$, including two regions representing the plates. The bottom plate is set as a stationary region, whereas the top plate is set to oscillate at a fixed frequency, and following the simple function,

$$\delta_x = \delta_{x,0}\sin(\omega t) \tag{21}$$

with $\delta_{x,0}$ defining the amplitude of the oscillation, and $\omega$ the frequency. The computational domain and lattice structure is presented in Figure 8.

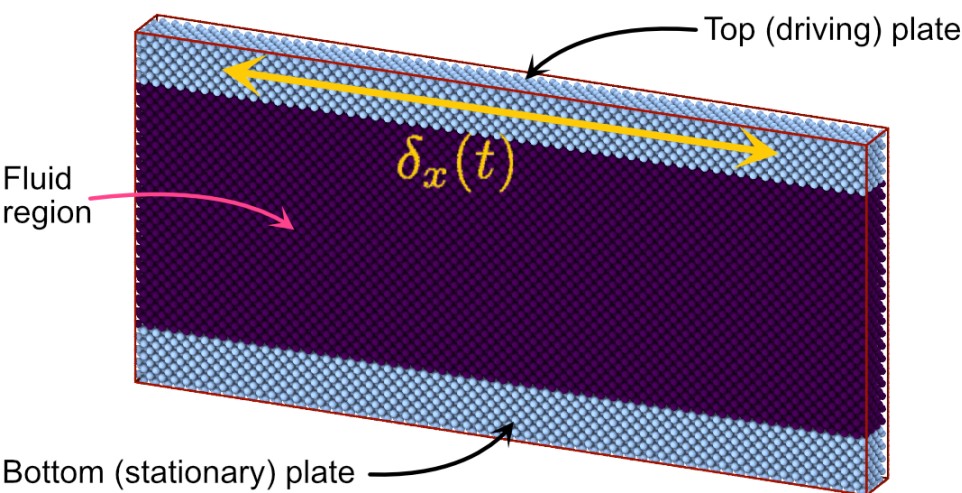

**Figure 8.** Configuration of oscillatory shear case.

Some preliminary tests were performed in order to gauge the ability of the proposed model to effectively reproduce, separately, the elastic or viscous response. A simple visual example of the test configuration and dynamic response for the pure elastic case is presented in Figure 9. In this figure a group of particles has been selected to be tracked as timeline and three different instants of the oscillatory test, for a pure elastic substance, are presented. Particles not included in the timeline group have been coloured using the velocity in the $x-$direction, which is the direction of the forced oscillating strain. The dynamic relationship between strain and stress for this case is illustrated in Figure 10, where the time evolution of both quantities are plotted and an in-phase evolution can be clearly observed. In Figure 11, are presented the results obtained for the Loss and Storage Moduli from the numerical tests using only the elastic component, for different values of the magnitude of $\epsilon_{CS}$, at different oscillatory shear amplitudes $\gamma$, and for three different oscillation frequencies $\omega$. As it can be appreciated, the model performs very well in defined ranges of shear strain amplitude, providing a response that exhibits a clear elastic behaviour. It is noteworthy, however, that some deviations from the expected response can also be observed. For instance, by examining Figure 11, it is always possible to distinguish a range of oscillation amplitudes where the Storage moduli is clearly larger than the Loss moduli, as predicted by theory for elastic materials and, although we did not explore a large number of frequencies, there is a clear frequency dependency, especially observable in the results for $\epsilon_{CS} = 1 \times 10^{-6}$ and $\epsilon_{CS} = 1 \times 10^{-5}$, as expected. Nevertheless, it is important to note that our model exhibits some numerical discrepancies at some of the extreme conditions explored and presented in Figure 11. Clearly, there is an operational range for the model and therefore simulations performed out of those bounds could produce unexpected behaviour, although this can be somehow anticipated. For example, if deformation magnitude is too small, the substance deformation is effectively masked by the simple relaxation of the lattice structure, and therefore not enough to transfer the stress to the neighbouring layers. This can be observed in Figure 11 where some of the explored cases failed to show a clear elastic behaviour at very low values of strain (i.e., at low values of oscillation amplitude). On the other hand, it is important to bear in mind that our proposed modelling approached is based on the concepts and theory of linear viscoelasticity, and therefore applicable mostly to small deformations. This is specially important from the elastic component perspective. It is then expected that the model fails at large values of strain, which can be observed also in Figure 11. In any case, the large range of values explored in the present work is somehow justified because although theory indicates that applicability of the linear viscoelastic behaviour should be expected at small deformations, there is not a clear sense of what "small deformation" might be in a generic substance, like those explored in this work. In fact, the range of parameters was selected only on the basis of exploring the

ability of our modelling approach to capture different substances, as well as the reliability and stability of our approach when employed under extreme conditions. Our results indicate that the applicability of our model requires an adequate selection of parameters and operating ranges to avoid problems with very large deformations, as well as with very small deformations, but that it is able to reliably reproduce expected theoretical elastic behaviour within appropriate ranges of shear strain.

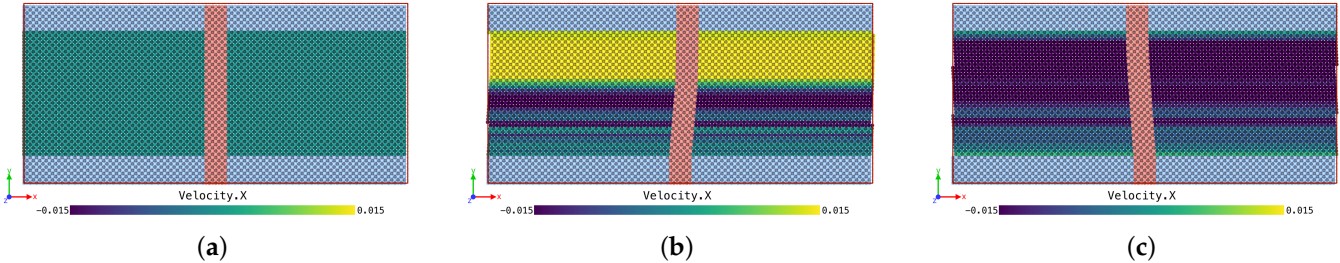

**Figure 9.** Illustration of set of particles tracked during the oscillatory test in a pure elastic substance. Particles not included in the timelines have been coloured by velocity in $x-$direction, at three different time instants. (**a**) $t = 0\,\mathrm{s}$. (**b**) $t = 5 \times 10^{-5}\,\mathrm{s}$. (**c**) $t = 4.5 \times 10^{-4}\,\mathrm{s}$.

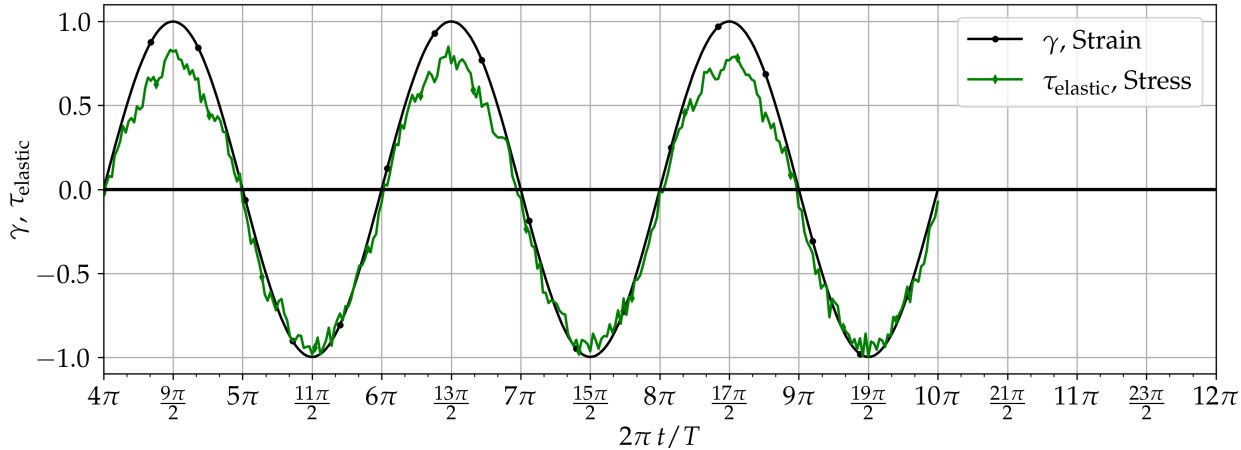

**Figure 10.** Shear stress $\tau$ and strain $\gamma$ vs. time in the example elastic case.

As our model involves a potential emulating the harmonic LSM model, it is important to asses the impact of oscillatory external forcing or deformation. Then, by performing external forced strain on a pure elastic substance, internal forces and stresses are developed, affecting the movement of each particle in the lattice. This can eventually cause harmonic instabilities to start and grow and, if care is not taken, these internal oscillations and instabilities can render the lattice and model inaccurate. For instance, from the numerical experiments, it was possible to observe that configurations with low level of stiffness caused the generation and propagation of internal longitudinal waves that, given the conditions, might disrupt the structure of the lattice, and therefore rendering the simulation unstable and not physically representative. Clearly, if the elastic potentials are set to low values, but above a given threshold, the behaviour is still elastic but with extremely low stiffness, which might still cause the appearance of some longitudinal waves. The important aspect to consider is then the numerical stability and physical consistency, which will help to determine a good configuration of parameters. Precisely, the parameters assignment for the repulsive component play also an important role in this stability. If the magnitude of repulsive potential, represented by $\epsilon_{\mathrm{Soft}}$, is made too small, internal oscillations and waves appear, and they might end up breaking the structure of the lattice. In the limiting stable cases, longitudinal waves appearing and propagating through the whole domain, bring about interference patterns that travel across the modelled substance, but without

causing disruption to the particles lattice. Examples of both situations are shown in Figures 12 and 13.

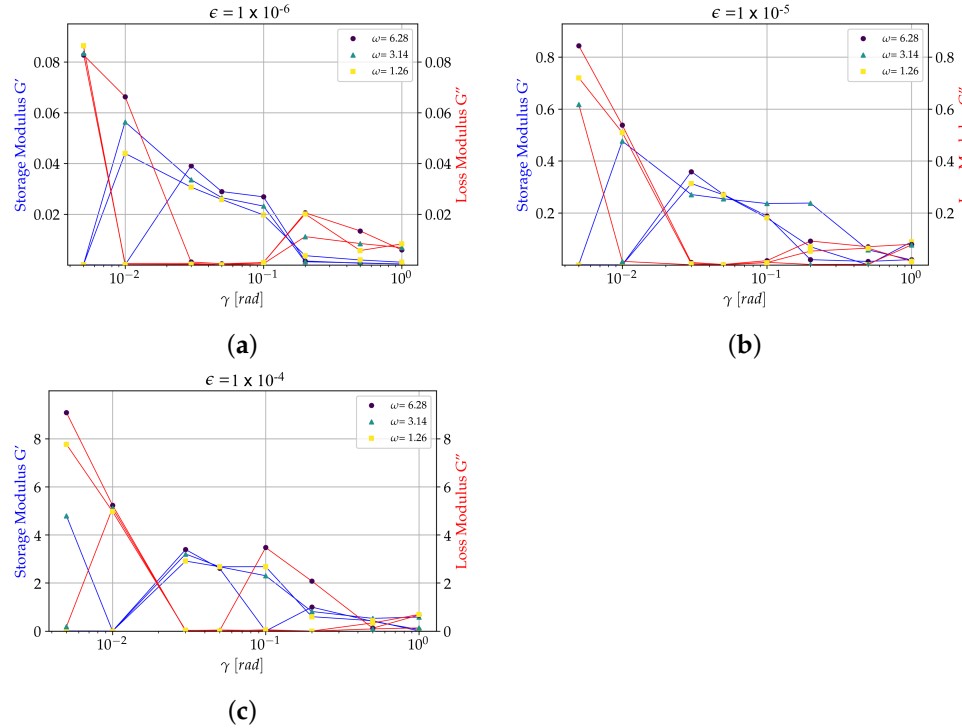

(a)

(b)

(c)

**Figure 11.** Storage and Loss moduli obtained with an exclusively elastic configuration of the proposed model. Blue: Storage modulus G′, red: Loss modulus G″. Tests obtained for $5 \times 10^{-3} \leq \gamma \leq 1$, and for oscillating frequencies $\omega = 1.26, 3.14, 6.28$ rad s$^{-1}$. (**a**) Results for $\epsilon_{CS} = 1 \times 10^{-6}$. (**b**) Results for $\epsilon_{CS} = 1 \times 10^{-5}$. (**c**) Results for $\epsilon_{CS} = 1 \times 10^{-4}$.

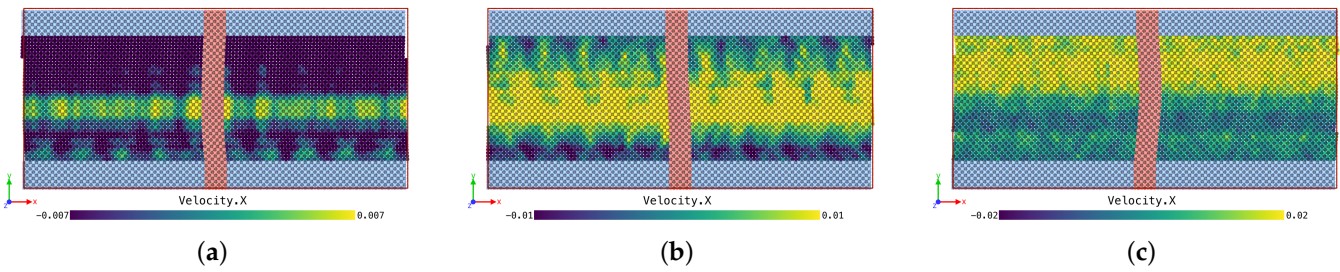

(a)

(b)

(c)

**Figure 12.** Set of particles tracked during the oscillatory test in a pure elastic substance. Example of a limiting stable case. Longitudinal waves below stability threshold. (**a**) $t = 3.9 \times 10^{-4}$ s. (**b**) $t = 4.3 \times 10^{-4}$ s. (**c**) $t = 7.8 \times 10^{-4}$ s.

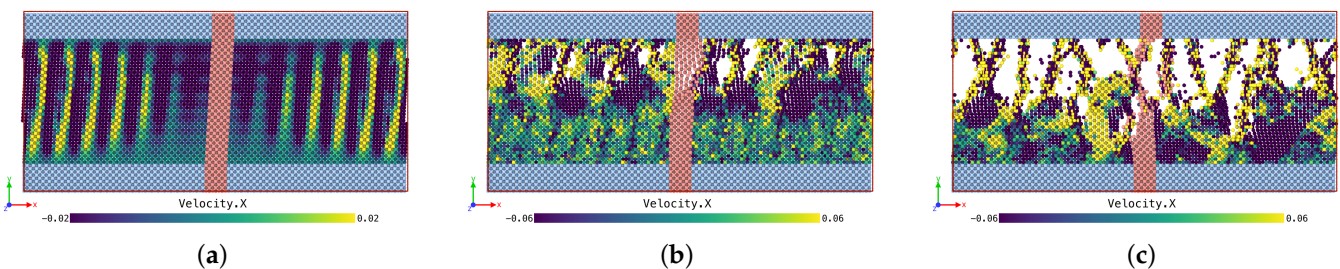

(a)

(b)

(c)

**Figure 13.** Set of particles tracked during the oscillatory test in a pure elastic substance. Example of a unstable case. Longitudinal waves above stability threshold. (**a**) $t = 5.5 \times 10^{-5}$ s. (**b**) $t = 7.5 \times 10^{-5}$ s. (**c**) $t = 8.5 \times 10^{-5}$ s.

The viscous component in our model is captured by a standard SPH model, as mentioned before. As it can be appreciated from Figures 14 and 15, the viscous component employed in our model is able to capture the intended dissipative nature, exhibiting the expected loss angle $\delta \approx 90°$ between the signals of strain and stress. The consistency of the viscous response was also tested for a number of nominal strain amplitudes and oscillation frequencies with the oscillatory strain test. Numerical results shown in Figure 16 allow us to conclude that the model is able to capture the expected behaviour of higher loss than storage moduli, for all the different frequencies and magnitudes of strain oscillation explored in the numerical experiments.

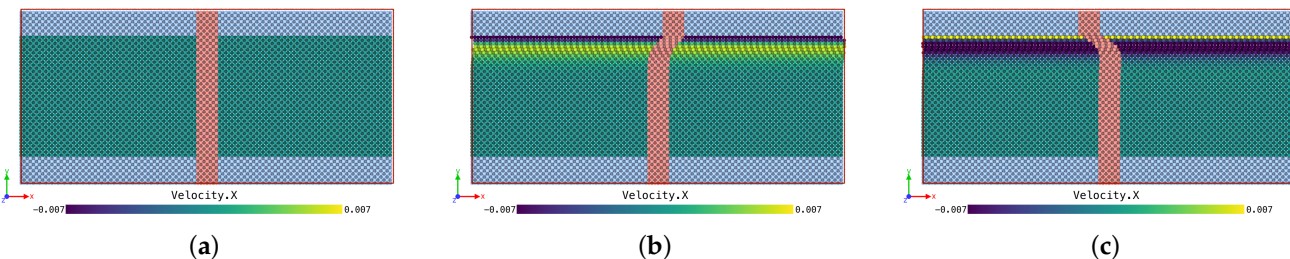

(a)                                (b)                                (c)

**Figure 14.** Set of particles tracked during the oscillatory test in a pure viscous substance. (**a**) $t = 0$ s. (**b**) $t = 1 \times 10^{-4}$ s. (**c**) $t = 4.7 \times 10^{-4}$ s.

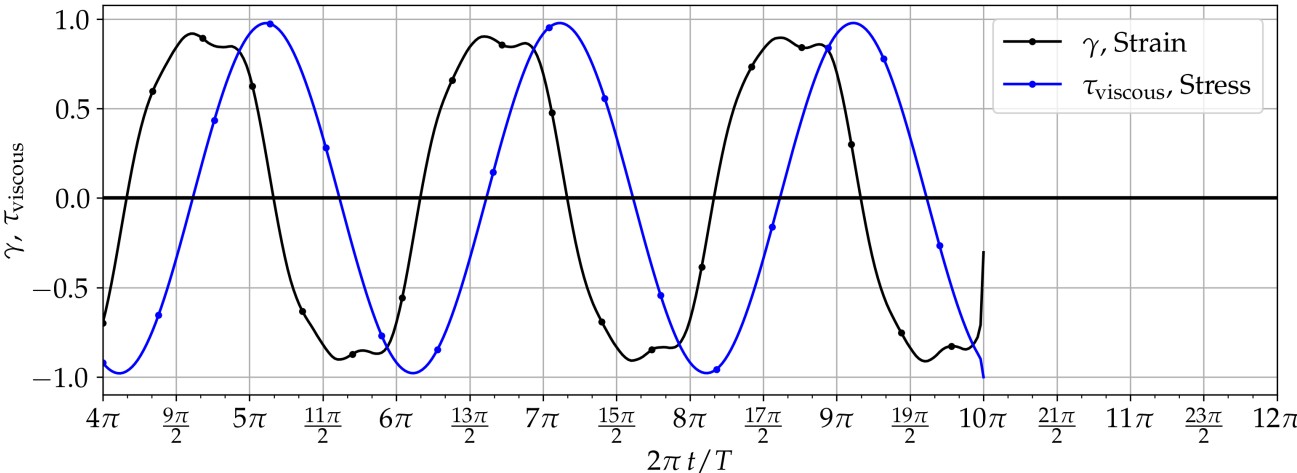

**Figure 15.** Shear stress $\tau$ and strain $\gamma$ vs. time in the example viscous case.

Results of the oscillatory tests obtained with the proposed viscoelastic methodology are presented graphically in Figure 17 for the timelines of a selected set of particles. Figure 17 shows the presence of elastic internal waves, similar to those longitudinal waves observed for the elastic configuration and presented in Figure 12. However, these internal waves are essentially below any unstable threshold, as they remain stable for more than 4 oscillation periods. The stability is also clear from the consistency of the lattice structure even after several periods of oscillation. In Figure 18, is shown the time evolution of shear stress ($\tau$) and strain ($\gamma$) for a selected viscoelastic configuration. It is clear that, compared to the elastic and viscous cases, the viscoelastic fluid shows a loss angle that is neither $\delta = 0°$ or $\delta = 90°$, which is very much in line with the results observed in some real viscoelastic substances that present stress and strain signals out of phase, but not to the degree of a fully viscous substance. This distinctive characteristic was tested for a number of oscillation frequencies and shear strain amplitudes. The results for the different shear conditions are presented in Figure 19. From the graphs it is clear that the response for the different viscoelastic settings are dependent from the oscillation frequency and the amount of strain imposed. This feature has been documented [56–58], and indeed it is one of the

main results of many experimental tests showing that substances response to shear exhibit a clear frequency dependency, as well as a strain magnitude dependency.

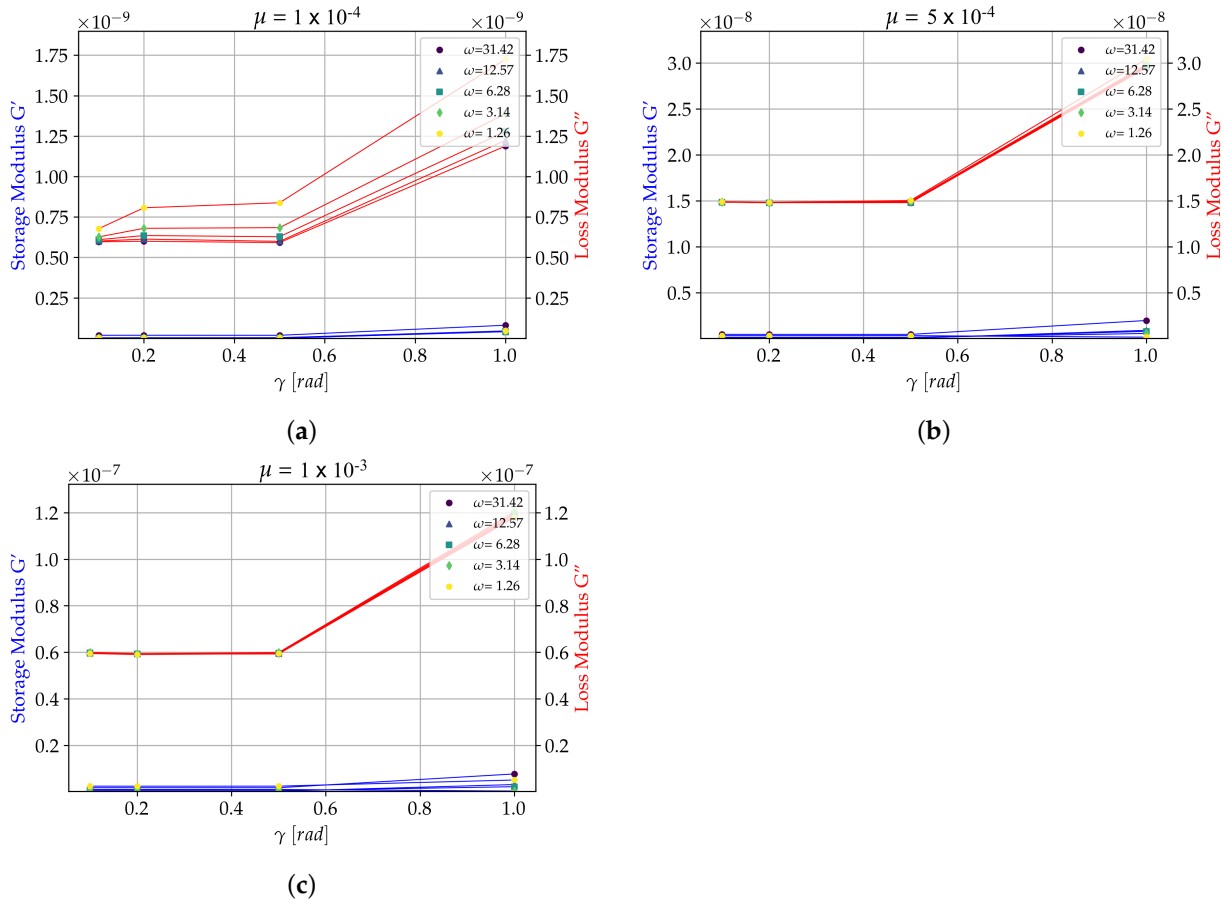

**Figure 16.** Storage and Loss moduli obtained with an exclusively viscous configuration of the proposed model. Blue: Storage modulus G′, red: Loss modulus G″. Tests obtained for $1 \times 10^{-1} \leq \gamma \leq 1$, and for oscillating frequencies $\omega = 1.26$, 3.14, 6.28, 12.57, 31.42 rad s$^{-1}$. (**a**) Results for $\mu = 1 \times 10^{-4}$. (**b**) Results for $\mu = 5 \times 10^{-4}$. (**c**) Results for $\mu = 1 \times 10^{-3}$.

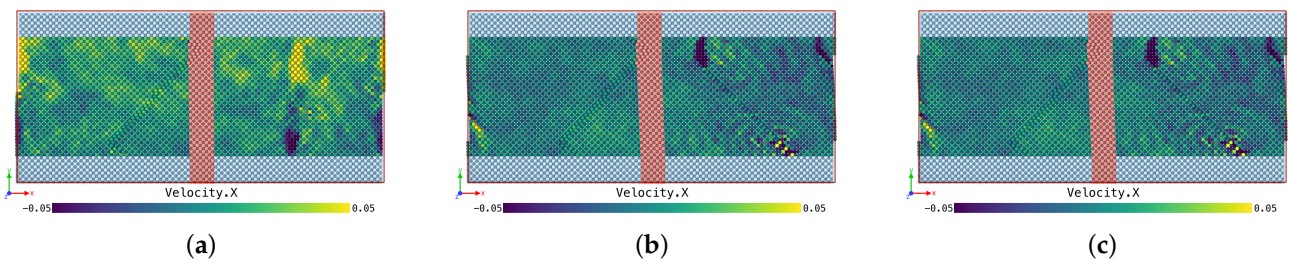

**Figure 17.** Illustration of set of particles tracked during the oscillatory test in a viscoelastic substance modelled with $\epsilon_{CS} = 1 \times 10^{-5}$ and $\mu = 0.1$. Particles coloured by velocity in $x-$direction, at three different time instants. (**a**) $t = 2.0$ s. (**b**) $t = 2.35$ s. (**c**) $t = 2.6$ s.

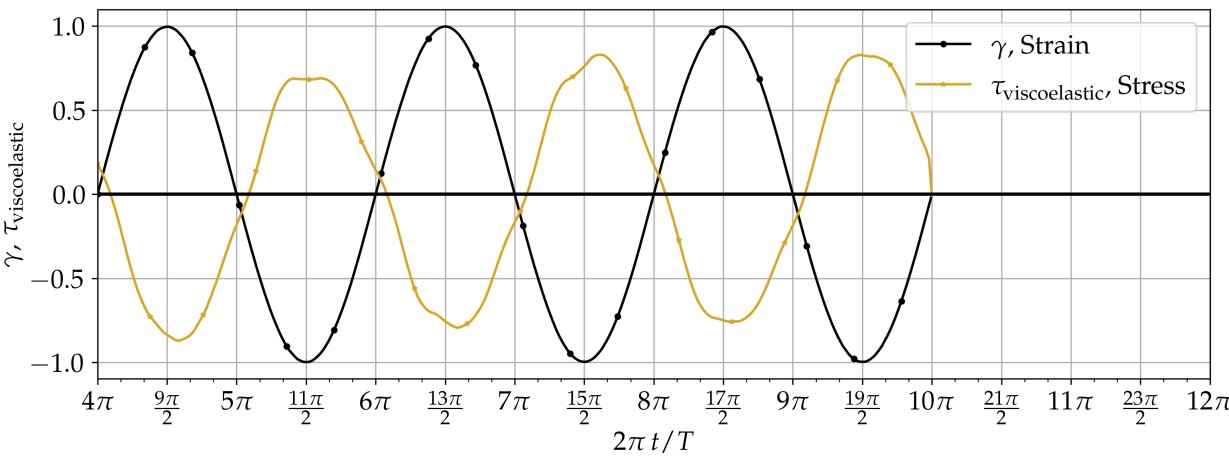

**Figure 18.** Shear stress $\tau$ and strain $\gamma$ vs. time for a viscoelastic case.

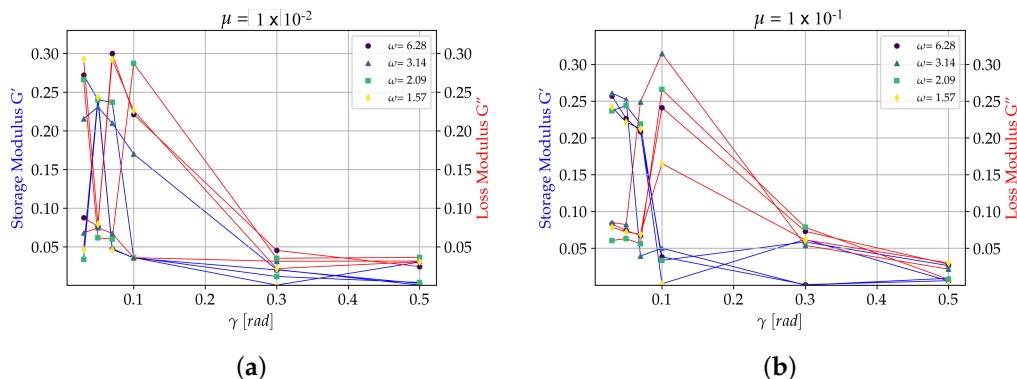

(**a**)                                          (**b**)

**Figure 19.** Storage and Loss moduli obtained for a combined viscoelastic configuration according to proposed model. Blue: Storage modulus $G'$, red: Loss modulus $G''$. Tests obtained for $3 \times 10^{-2} \leq \gamma \leq 0.5$, and for oscillating frequencies $\omega = 1.57, 2.09, 3.14, 6.28$ rad s$^{-1}$. (**a**) Results for $\mu = 1 \times 10^{-2}$. (**b**) Results for $\mu = 1 \times 10^{-1}$.

## 5.2. Viscoelastic Flows in Cylindrical Pipes

In the previous section, we showed that the model can produce materials whose storage and loss moduli are consistent with those of viscoelastic materials. However, this was the result of very small fluctuations that do not generate a real flow in the material. In this section, we will discuss the case of pipe flow, which produce a fluid-like motion of the particles and it is one of the most common mechanisms of transport of fluid-like substances. This particular flow, from the perspective of a viscoelastic substance, is also extremely important as it represents a standard constant shear condition employed in many viscometers. In this type of flow, a well-known characteristic is the expected velocity profile, specially for laminar low speed flows. In the case of a mostly viscous or Newtonian substance, both experience and theory have demonstrated that a parabolic velocity profile is formed, with maximum or peak velocity present in the centre-line of the pipe. Instead, for a mostly elastic fluid, the viscoelastic nature produces the so-called Bingham velocity profile or Bingham flow, where the velocity remains mostly constant in the cross section within the pipe, except near the pipe walls where there is usually a sharp drop towards the no-slip velocity at the wall. This type of flow is usually also known as "plug flow". The Bingham fluids are characterised by the existence of a yield stress and their ability to also transmit a shear stress without a velocity gradient, unlike Newtonian fluids. Nevertheless, in order to make Bingham fluids flow, the driving shear stress has to be larger than the yield stress. Below this yield stress the fluid will behave almost like a solid body and above as a liquid. Interestingly, although a precise estimation of the nature of the viscoelastic

flow will require the estimation of such a yield stress, through the examination of the flow velocity profiles it is possible to ascertain what the respective shear stress is present in a given flow. Specifically, the ability of our model to capture viscoelastic behaviour under constant shear was explored in a constant gradient pipe flow case. Before presenting the simulation setup and results, a brief summary of the models and relationships to estimate some characteristics of a Bingham flow are included next.

### 5.2.1. Bingham Flows: Velocity Profiles and Yield Stress

As the numerical experiments presented in this section were performed in a cylindrical domain, it is convenient to express the shear stress in cylindrical coordinates and for a general condition (Newtonian or Bingham). For this, we consider an incompressible, laminar flow under the effects of a general pressure gradient, that in general can be decomposed into a gravity component ($f_x$) and a standard pressure difference $\Delta P$, in a system of length $L$, which might be at an angle $\beta$ to the vertical. Neglecting end effects, by assuming the dimension of the system in the radial direction is relatively small compared to that in the axial direction ($L$), and assuming an axial flow so $v_r = 0$, $v_\theta = 0$, and $v_z \neq 0$, it is possible to obtain a general expression for the momentum conservation equation as,

$$- \mu \nabla^2 v_z = -\nabla p + \rho g_z \tag{22}$$

where $\mu$ is the dynamic viscosity, $\rho$ the density of the flow, and $v_z$ the velocity along the axis of the pipe. In this expression it is also assumed that there are small flow rates so that the viscous forces impose strictly uniform flow. With this assumption $v_z$ is independent of $z$ and we may reasonably postulate that the velocity $v_z = v_z(r)$ and pressure $p = p(z)$. In this manner, the only non-vanishing components of the stress tensor are $\tau_{rz} = \tau_{zr}$, which depend only on $r$, and which can then be expressed as,

$$\tau_{rz} = \frac{\Delta P}{2L} r \tag{23}$$

These equations are derived without making any assumption about the type of fluid and so are applicable to both Newtonian and non-Newtonian fluids. If we use Newton's law of viscosity, $\tau_{rz} = -\mu \frac{d}{dr} v_z$ we can use Equation (23) to generate a differential equation for the velocity, which after integration gives the following flow profile,

$$v_z = -\frac{\Delta p}{4\mu L} r^2 - \frac{C_1}{\mu} \ln(r) + C_2 \tag{24}$$

that, however, is only applicable to Newtonian fluids due to the use of Newton's law. To construct the flow velocity for Bingham fluids, we must make the following considerations: (i) The velocity profile of Newtonian fluids in pipe flows consists of a velocity gradient which decreases towards the centre of the pipe which in turn causes the shear stress, transmitted by fluid layers, to decline toward the pipe centre. (ii) Since Bingham fluids become solid when the applied shear stress falls below the yield stress we recognize that Bingham-fluids will become solid in the central layers of the pipe. Thus, we will have a solid 'plug' moving within the flow. (iii) In the process of deriving the velocity profile the radius of this solid area has to be additionally determined. Using these considerations, and applying adequate boundary conditions, it is possible to obtain a simple Bingham model where there is no flow until the critical/yield stress $\tau_0$ is reached:

$$\eta \to \infty \ \text{ or } \ \frac{d}{dr}(v_z) = 0 \qquad \text{when } |\tau_{rz}| \leq \tau_0 \tag{25}$$

$$\eta = \mu_0 + \frac{\tau_0}{\pm \frac{d}{dr}(v_z)} \ \text{ or } \ \tau_{rz} = -\mu_0 \frac{d}{dr}(v_z) \pm \tau_0 \qquad \text{when } |\tau_{rz}| \geq \tau_0 \tag{26}$$

where $\eta$ is the non-Newtonian viscosity and $\mu_0$ is a Bingham model parameter with units of viscosity. $\tau_{rz}$ is positive when the positive sign is used with $\tau_0$ and the negative sign with $d(v_z)/dr$. We can calculate the yield stress $\tau_0$ by considering that $\tau_{rz} = \tau_0$ at some $r = r_0$. This follows from both Equations (25) and (26) at $|\tau_{rz}| = \tau_0$, which gives the following expression for the yield stress:

$$\tau_0 = \frac{\Delta P}{2L} r_0 \tag{27}$$

It is clear that the velocity profile can be split it into an inner ($r \leq r_0$) and outer ($r_0 \leq r \leq R$) region, where appropriate models for these profiles are:

$$v_{zo} = \frac{\Delta P}{4\mu_0 L} R^2 \left(1 - \frac{r^2}{R^2}\right) - \frac{\tau_0}{\mu_0} R \left(1 - \frac{r}{R}\right) \qquad \text{for } r_0 \leq r \leq R \tag{28}$$

for the flow velocity profile in the outer region, and

$$v_{zi} = \frac{\Delta P}{4\mu_0 L} R^2 \left(1 - \frac{r_0}{R}\right)^2 \qquad \text{for } r \leq r_0 \tag{29}$$

for the inner region. This last model gives a constant velocity in the inner region, as expected, for a fluid with plug flow, with $r_0 = 2L\tau_0/\Delta P$ being the radius of the plug-flow region in the central part of the pipe. In this manner, the velocity profile is parabolic in the outer region as given by Equation (28) and is flat in the inner region as given by Equation (29). Finally, the mass flow rate can be obtained by integration by parts of the velocity profile over the cross section of the circular pipe,

$$\dot{m} = \frac{\pi R^3 \rho}{\tau_R^3} \int_0^{\tau_R} \tau_{rz}^2 \left(-\frac{d}{dr}(v_z)\right) d\tau_{rz} \tag{30}$$

where we have used Equation (23) so that $r/R = \tau_{rz}/\tau_R$, with $\tau_R = (\Delta P/(2L))R$ defining the shear stress at the wall. After some manipulation, it is possible to obtain the so-called Buckingham–Reiner equation for the mass flow rate,

$$\dot{m} = \frac{\pi \Delta P R^4 \rho}{8\mu_0 L} \left(1 - \frac{4}{3}\frac{\tau_0}{\tau_R} + \frac{1}{3}\frac{\tau_0^4}{\tau_R^4}\right) \tag{31}$$

where the mass flow rate is defined in terms of the yield stress $\tau_0$ and the wall shear stress $\tau_R$. Note that no flow occurs below the yield stress, so the equation is valid only for $\tau_R > \tau_0$.

### 5.2.2. Pipe Flow-Numerical Experiments

In order to test the ability of our model to capture viscoelastic behaviour at different spatial ranges, numerical experiments on small- and medium-scale pipes were performed for a number of parameters adequate to our model. For instance, the small-scale simulations were performed in a pipe with inner radius $r_{\text{inner}} = 2 \times 10^{-3}$ m and length $L = 2.4 \times 10^{-2}$ m. The SPH simulation was configured with a FCC lattice cubic of size $\Delta_L = 2.5 \times 10^{-4}$, and smoothing length defined as $h = 1.95\,\Delta_L$. A schematic representation of the pipe geometry used is presented in Figure 20. Experiments were performed for a number of values of our base potential elastic model $\epsilon_{\text{CS}}$, and using the viscous SPH Taitwater–Morris model with a set of values for the SPH dynamic viscosity. The combination of these parameters allowed us to reproduced a range of viscoelastic behaviours in response to a body force imposed over the set of fluid particles. The virtual fluid was forced to flow by imposing a body force $f_x$ in the $x-$direction, while the domain was defined as periodic along the flow direction.

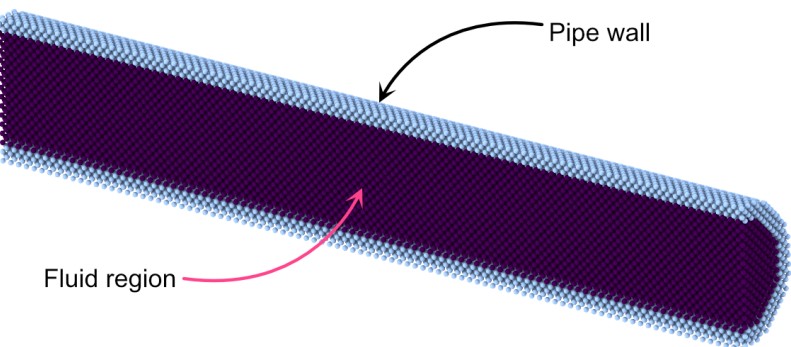

**Figure 20.** Pipe geometry for the small scale simulations.

In order to visualize the multiple flow regimes obtained, both velocity profiles and snapshots of the particles of our "numerical fluid" at different times have been produced. For the latter, a timeline analysis, a set of particles is preselected to represent a control volume travelling with the particles at different time instants, in the same fashion as in the oscillatory tests. The selection of particles used to construct the timelines is shown in Figure 21 as observed at the initial time, $t = 0$ s.

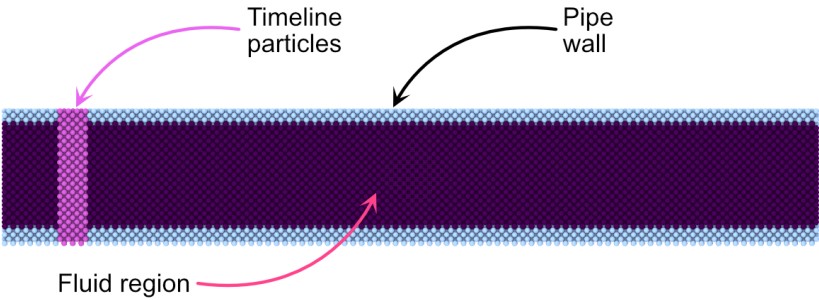

**Figure 21.** Tracking particles for construction of timelines in pipe simulations.

The model was assessed by a parametric study changing 3 variables: the elastic factor $\epsilon_{CS}$, the driving body force $f_x$, and the dynamic viscosity $\mu$. It is worth noting that, an additional parameter in our model that is tied to the factor $\epsilon_{CS}$, is the factor of the repulsive potential $\epsilon_{Soft}$. As it was described earlier, this parameter was linked to the $\epsilon_{CS}$ by a proportional relation, although for this case a reasonably dependency was found to be given as $\epsilon_{Soft} = 0.4\epsilon_{CS}$. Varying the coefficients in turn means that we are both varying the attractive and the repulsive potentials, therefore changing the ratio between elastic and viscous nature.

From the experiments, the cases exhibiting a Newtonian flow featured the expected predominant parabolic shape. The resolution of the velocity profiles was constrained to to the limit of particle points in the simulation, which caused a few Newtonian cases to lack a well defined parabolic shape and would tend to slightly skew from this pattern when approaching the wall, as it can be seen in Figure 22 below for the Newtonian plot. On the same figure the results for a Bingham flow are also presented. This flow, presented with a blue line, was almost ideal in shape, with the plug flow region containing mainly zero velocity gradient (as expected). As before there are some points that skew from the theoretical shape, specially at the yield stress point, where it can be seen that the flow increases locally before behaving Newtonian as one approaches the wall. Another case presented in Figure 22 is a flow that displays a mixed behaviour. For this generic flow, we see characteristics of Newtonian behaviour as well as features of a Bingham plug flow.

However, the regions in which the plug flow occurs do not follow the theory for Bingham flows, so it cannot be considered properly Bingham. This is because the flow is only plug flow in two regions: at a coaxial annulus and at a smaller radius around the centre, whilst maintaining Newtonian flow features elsewhere.

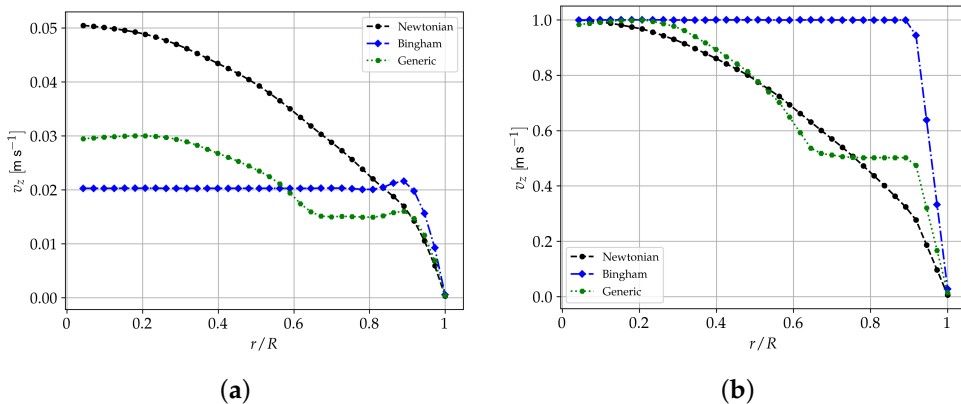

**Figure 22.** (**a**) Velocity profiles in axial direction in the pipe flow. profiles obtained for Newtonian, Bingham and Generic cases. (**b**) Normalised axial velocity profiles.

For the timelines analysis, we first considered the case when the attractive, and hence the repulsive potentials, were null by assigning $\epsilon_{CS} = 0$. As established earlier, these represent an elastic potential when considered together, and hence we expect a Bingham plug flow when this potential is dominant. On the other hand, for conditions where $\epsilon_{CS} = 0$ we expected flows to be essentially viscous Newtonian in nature. Our model is able to capture this condition, as ratified by the results presented in Figure 23, where the flow can be regarded as laminar, but more importantly Newtonian. Clearly, the flow travels downstream and, as the domain is periodic in the $x-$direction, particles re-enter the domain in the left side of the pipe still maintaining the parabolic profile, as seen in the third time snapshot at $t = 0.4032$ s in this figure.

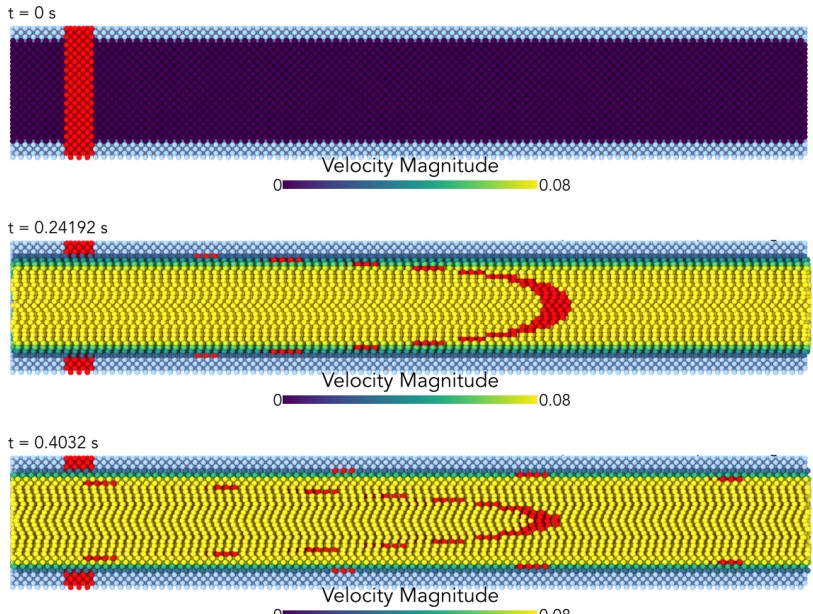

**Figure 23.** Snapshots of timelines evolution (tracking particles) in a pipe for a Newtonian viscous flow, obtained at three different instants. At third instant particles are going through the domain for the second time, as per periodic configuration. Flow obtained with: $\epsilon_{CS} = 0$, $\rho = 1 \times 10^3$, $c = 1 \times 10^{-1}$, $\mu = 1 \times 10^{-3}$, $g = 5 \times 10^{-1}$.

By assigning $\epsilon_{\text{CS}}$ to any non-zero value, it is expected that the model shows a mixed behaviour between complete Newtonian or complete Bingham, being the latter the extreme case of our viscoelastic model. By just setting $\epsilon_{\text{CS}} = 1 \times 10^{-14}$, even with a relatively low viscosity of $\mu = 1 \times 10^{-3}$, the model brings about a flow that resembles more a Bingham plug flow, as shown in Figure 24. However, some caution must be advised when assuming only these parameters to be involved in determining the flow type. This can be seen by the next result in which the viscosity was reduced to 3 orders of magnitude below, and we obtained Bingham flow, even with no elastic potential in action Figure 25.

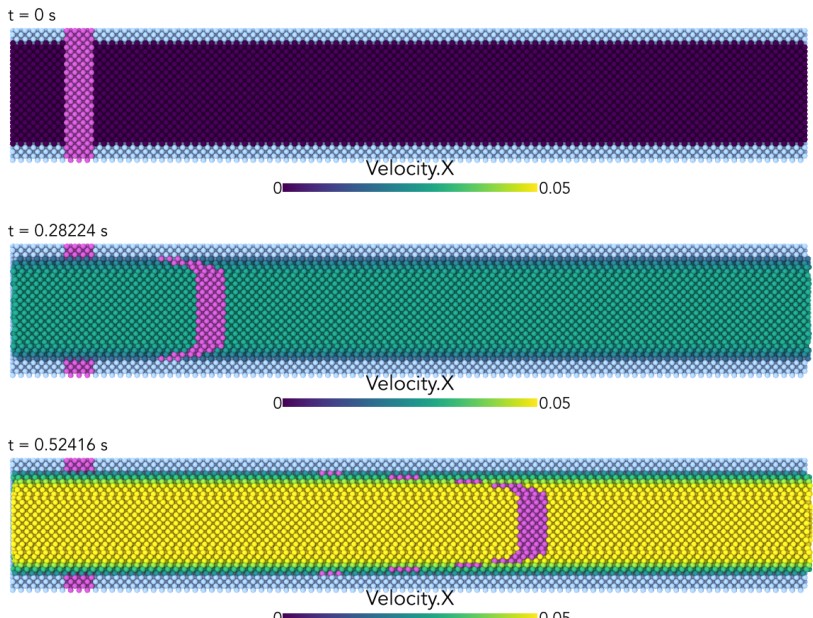

**Figure 24.** Evolution of timelines (tracking particles) in the pipe for response type "Bingham flow". Flow obtained with: $\epsilon_{\text{CS}} = 1 \times 10^{-14}$, $\rho = 1 \times 10^3$, $c = 1 \times 10^{-1}$, $\mu = 1 \times 10^{-4}$, $g = 1 \times 10^{-1}$.

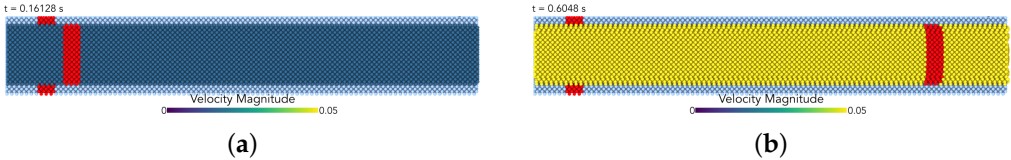

**Figure 25.** Bingham flow obtained with null elastic component. (**a**) Flow timelines at t = 0.16 s. (**b**) Flow timelines at t = 0.6 s. Flow obtained with: $\epsilon_{\text{CS}} = 0$, $\rho = 1 \times 10^3$, $c = 1 \times 10^{-1}$, $\mu = 1 \times 10^{-6}$, $g = 1 \times 10^{-1}$.

For conditions of constant shear as in the pipe flow, the elastic factor $\epsilon_{\text{CS}}$ is still a variable that plays a role into the possible appearance of some instabilities, as could be expected from the results of the oscillatory shear case. The change of settings in the model, from a case with $\epsilon_{\text{CS}} = 0.0$ to a configuration with $\epsilon_{\text{CS}} = 1 \times 10^{-12}$ produces a dramatic alteration of the behaviour of the flow, as appreciated in Figure 26. At $t = 0.16$ s the slow speed plug flow that was just beginning to form with $\epsilon_{\text{CS}} = 0.0$ (see Figure 25), has already developed an abnormal pattern, with the lattice starting to be disrupted, and some regions of low and high velocity appearing in an alternating fashion in the radial direction. The instability finally produces a completely ill-conditioned lattice, with large scale voids, that are completely clear at $t = 0.4$ s, in contrast with the completely stable plug flow obtained with $\epsilon_{\text{CS}} = 0.0$, that even at $t = 0.6$ s keeps a essentially flat velocity profile towards the centre of the pipe. In any case, once a set of parameters is identified as a potential troublesome setting, by a simple re-tuning of the model it is possible to effectively subdue the detected unstable mode. In spite of those few cases developing instabilities, the

model behaves extremely well, considering the breadth of potential conditions that can be captured with the proposed methodology.

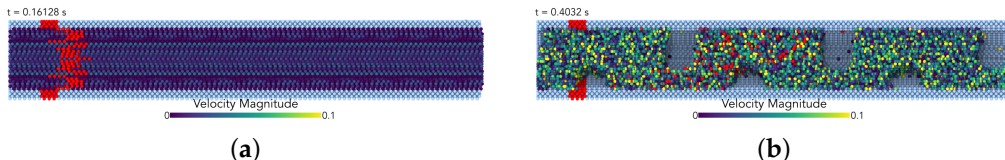

**Figure 26.** Unstable Bingham flow obtained with $\epsilon_{CS} = 1 \times 10^{-12}$. (**a**) Timelines at t = 0.16 s after the start of the simulation. (**b**) Timelines at t = 0.40 s. Flow obtained with: $\epsilon_{CS} = 1 \times 10^{-12}$, $\rho = 1 \times 10^3$, $c = 1 \times 10^{-1}$, $\mu = 1 \times 10^{-6}$, $g = 1 \times 10^{-1}$.

To analyse the behaviour of the flow with respect to these variables more rigidly, we propose a dimensionless constant $\pi_1$ defined as,

$$\pi_1 = \frac{g\rho^2 h^7}{\mu^2} \tag{32}$$

By constructing graphs of $\pi_1$ vs. $\mu$ for the different numerical experiments, it is possible to discern an operating region where the flow exhibited a "Bingham flow" response, as presented in Figures 27 and 28. In those plots we have coloured those points that exhibited predominantly plug flow. As can be seen from these figures, at higher values of $\pi_1$, or equivalently at lower values of $\mu$, our model brings about a mostly viscoelastic behaviour, as would be expected. From the graphs it seems clear that there is a consistency in the region for which $\pi_1$ relates to Bingham flow. As a simple preliminary guide, values of yield stress obtained in the numerical experiments are plotted against $\pi_1$ for a number of values of $\epsilon_{CS}$ in Figure 29. From this graph, it is clear that yield stress can be obtained at low levels of $\pi_1$, but only if the elastic factor $\epsilon_{CS}$ is high enough. It is also important to highlight that large values of $\tau_0$ can be obtained with our model even at low levels of stiffness, as there are some points showing high yield stress, even though the elastic factor was relatively reduced in comparison. Noteworthy, although there is an apparent region in the plot where the values of yield stress are clustered, it is clearly necessary to perform additional experiments to corroborate the full validity and application of the region observed in Figure 29.

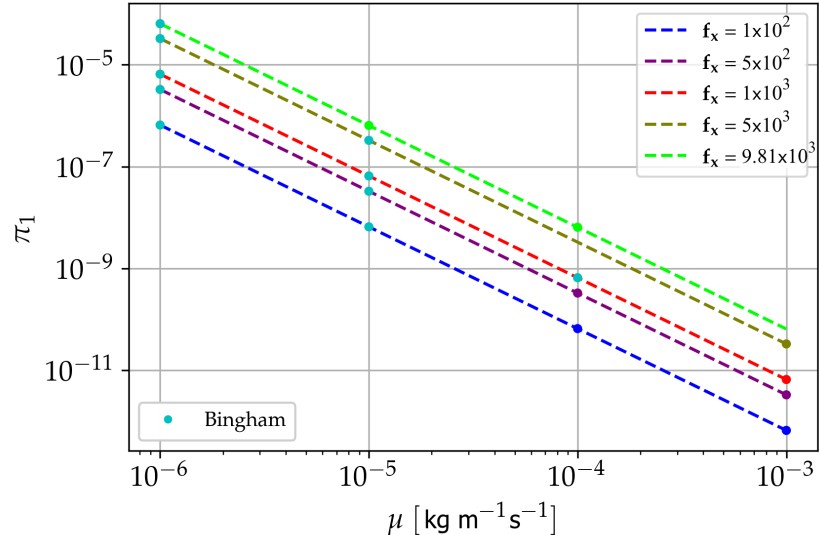

**Figure 27.** Dimensionless constant $\pi_1$ vs. $\mu$ at $\epsilon_{CS} = 0$ for different values of $f_x$.

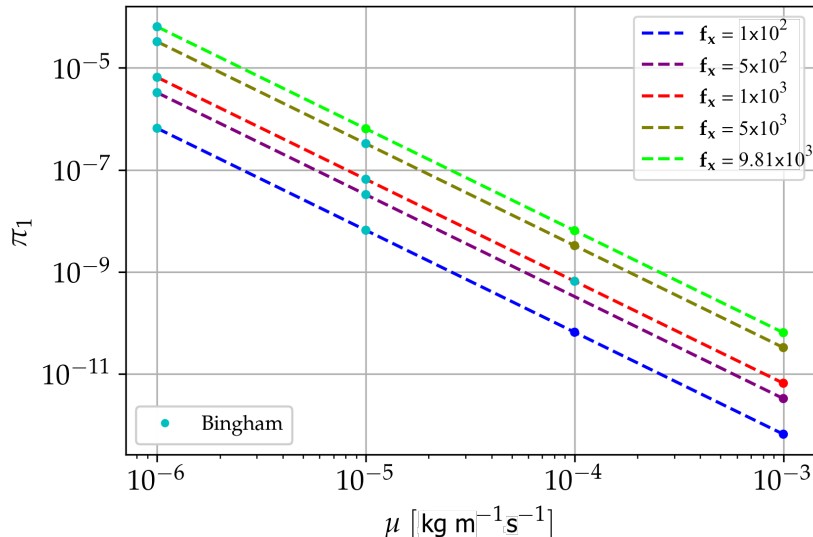

**Figure 28.** Dimensionless constant $\pi_1$ vs. $\mu$ at $\epsilon_{CS} = 1e - 14$ for different values of $f_x$.

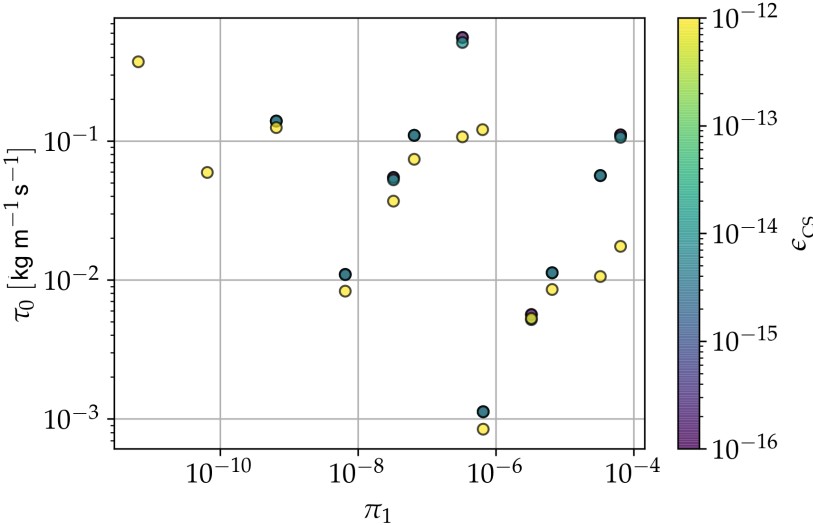

**Figure 29.** Values of yield shear stress $\tau_0$ obtained in the pipe flow numerical experiments for different values of $\pi_1$ and $\epsilon_{CS}$.

### 5.3. Column Collapse Due to Gravitational Potential

A third set of experiments was devised aiming to get a general overview of the effect of overlaying the potentials as proposed in this work in a more general and everyday situation. A benchmark case employed to show the ability of SPH to capture, among other phenomena, the free surface dynamics of a substance flowing freely is that of a two-dimensional column of water that suddenly collapses due to a gravitational force (see [59–61]). We test the hybrid modelling in two similar alternative cases, i.e., following the same rationale of the liquid column collapse, although with two volumetric domains of substance: a square prismatic column, and a cylindrical column, as shown in Figure 30. In these experiments an initial column of substance is enclosed within a cubic box, indicated partially by the dark grey particles in Figure 30 and used as container with non-permeable walls. The substance is then subjected to a vertical body force (in the form of an acceleration), for instance representing a gravity field in the negative $z - direction$. The box was prescribed as a cube with an internal volume of 1 m × 1 m × 1 m. The column of substance was prescribed in one of the experiments as a square prism with dimensions 0.5 m × 0.5 m × 0.9 m, in the $x-$, $y-$ and $z-$directions, respectively. In the second exper-

iment the substance column, also contained within a box of same dimensions as in the previous experiment, was prescribed as a cylinder with a height of 0.9 m, and a base radius of 0.25 m. In both cases gravity was set to 9.81 m/s$^2$ and density was set to a standard density of $1 \times 10^3$ kg/m$^3$.

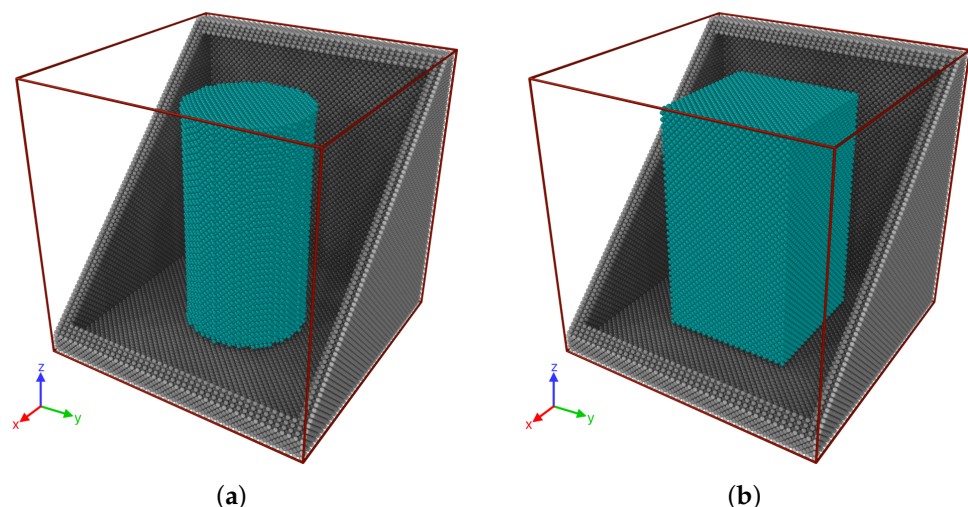

(a)                                                              (b)

**Figure 30.** General view of particles configuration for the substance column collapse. Wall particles shown in dark-grey; substance particles shown in light blue. Cutting plane only for visualisation. (**a**) Cylindrical column. (**b**) Prismatic column.

Numerical experiments were performed using a similar SPH configuration as previously described, with a smoothing length h = 0.0525 m, and an initial face-centred cubic lattice (`fcc`). This set of experiments was performed using the SPH model as proposed by [44]. The selected SPH model adopted for these tests aimed at exploiting its ability to simulate flows at relatively high velocities with a consistent numerical stability. As presented previously, and discussed extensively by other authors (e.g., [45,52]), the Monaghan's model uses an artificial viscosity that can easily be converted to the standard absolute viscosity thanks to a widely accepted equivalence between the real dynamic viscosity $\mu$ and the dissipation factor $\alpha$. The relationship between $\alpha$ and $\mu$, presented in Equation (17), was employed here, with a minor modification to account for the effect of the imposed body force. Equally, with the goal of discriminating between the different cases explored numerically, a second non-dimensional relationship was constructed using the main parameters of our model, although tailored to the case of a column of a substance collapsing by its own weight, and given as

$$\pi_2 = \frac{\mu\, h\, c^3}{g\, \epsilon_{\mathrm{CS}}} \qquad (33)$$

where $\mu$ is the dynamic viscosity, $c$ the numerical speed of sound, $h$ the smoothing length and $\epsilon_{\mathrm{CS}}$ the factor of the attractive component of the equivalent elastic potential. As usual, the value of the factor for the repulsive potential $\epsilon_{\mathrm{Soft}}$ has been intentionally omitted but defined in terms of $\epsilon_{\mathrm{CS}}$. Numerical experiments showed that in this case the model was not strongly affected by variations of $\epsilon_{\mathrm{Soft}}$ and that changes of its value, in a given range, had minimal influence on the overall performance and prediction capabilities. For instance, at the length and time scales involved in this case, the inclusion of the repulsive potential showed consistent numerical stability for values of potential magnitude in a range between 20% and 60% of the magnitude of the attractive potential. Figure 31, suggests that changing the ratio from 0.2 to 0.6 has a minimal impact in the overall evolution of the collapse of the column of liquid simulated. This appreciation is reinforced when examining the energy budgets presented in Figure 32, where the internal, kinetic, potential, and total

energy components are plotted in time. The local maxima and minima for the kinetic and internal potential energies occur essentially at the same instants, and their values are also equivalent, making even difficult to distinguish the energy budgets evolution from each other. The remaining tests presented through the rest of the present section use $\epsilon_{\text{Soft}} = 0.4\,\epsilon_{\text{CS}}$, unless otherwise stated.

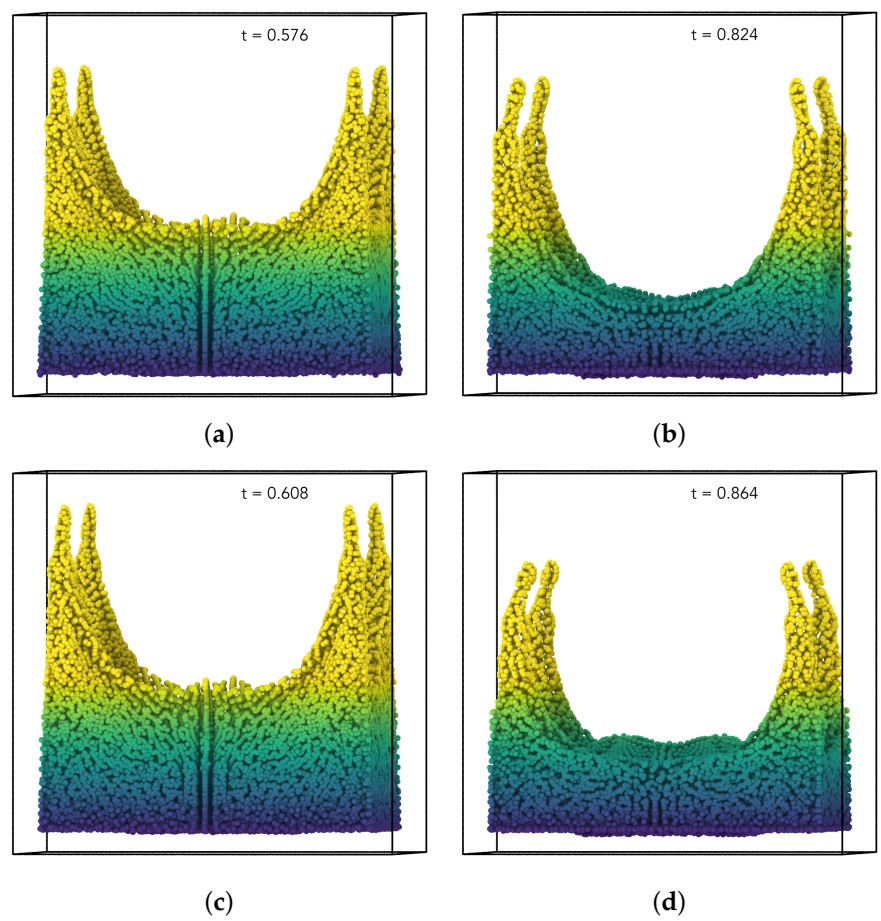

**Figure 31.** Column collapse at two different instants for different $\epsilon_{\text{Soft}}/\epsilon_{\text{CS}}$ ratios. Experiments performed for $\pi_2 = 9.9 \times 10^2$. (**a**) $\epsilon_{\text{Soft}} = 0.2\epsilon_{\text{CS}}$, $t \approx 0.4\tau$ ($\tau = 1.50$ s). (**b**) $\epsilon_{\text{Soft}} = 0.2\epsilon_{\text{CS}}$, $t \approx 0.55\tau$ ($\tau = 1.50$ s). (**c**) $\epsilon_{\text{Soft}} = 0.6\epsilon_{\text{CS}}$, $t \approx 0.4\tau$ ($\tau = 1.57$ s). (**d**) $\epsilon_{\text{Soft}} = 0.6\epsilon_{\text{CS}}$, $t \approx 0.55\tau$ ($\tau = 1.57$ s).

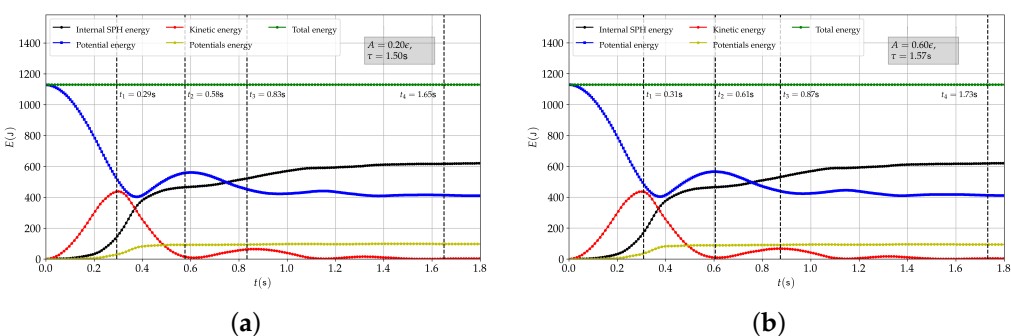

**Figure 32.** Time evolution of the components of the energy budget for the column collapse case. Viscoelastic response obtained by using $\epsilon_{\text{CS}} = 1 \times 10^{-4}$ and two values for $\epsilon_{\text{Soft}}$. Simulation settings: $\mu = 1 \times 10^{-3}$, $c = 10.0$, $\nu_{\text{artf}} = 0.8$, $\rho = 1 \times 10^3$, $h = 5.25 \times 10^{-2}$, $\Delta_L = 2.5 \times 10^{-2}$, $\pi_2 = 9.9 \times 10^2$. (**a**) $\epsilon_{\text{Soft}} = 0.2\epsilon_{\text{CS}}$. (**b**) $\epsilon_{\text{Soft}} = 0.6\epsilon_{\text{CS}}$.

An example of the effect of changing or tuning the equivalent quasi-elastic potential, is presented in Figure 33, where a substance with the same viscous component $\mu$, and

under the effect of the same vertical downwards acceleration of $g_z = -9.81$, develop three different collapse evolutions in time. For these cases, by examining the evolution of the energy budget, it was possible to assess the impact of modulating the viscoelastic response through several values of the quasi-elastic potential. As in Figure 32, Energy vs. time plots reflecting on the evolution of the main components of the energy budget are used to assess the impact of altering the elastic factor $\epsilon_{CS}$. In the energy budget plots, a simple characteristic time was employed to analyse the results at different instants. This characteristic time was defined as the point in time when the internal SPH energy reaches 99% of its steady-state value,

$$\tau = t_{e_{\mathrm{sph}}=0.99e_{\mathrm{SS}}}$$

Results for three different elastic factors are presented here: $\epsilon_{CS} = 1 \times 10^{-2}, 1 \times 10^{-3}$, $1 \times 10^{-4}$. In the first case, for $\epsilon_{CS} = 1 \times 10^{-2}$, Figure 33a–c show a extremely slow collapse of the substance column, clearly mimicking a gel-like behaviour, where the substance is affected by the downwards acceleration, but not really collapsing completely. Even after $t = 0.464$ s, when the collapse has already stopped and a balanced steady state has been reached, the column of the substance is still distinguishable. The time elapsed until steady state is reached has been estimated as $t \approx 0.43$ s, based on the time-story of the energy budget shown in Figure 34. On the other hand, Figure 33g–i show the evolution of the column collapse for a substance with $\epsilon_{CS} = 1 \times 10^{-4}$, that clearly presents a complete fluid-like behaviour. The collapse lasts $t = 1.76$ s, but during this time there is even a clear formation of a substance bouncing stage, observable at $t = 0.896$ s. The probable formation of ripples, and the occurrence of more than one bounce, can be inferred from the energy budget curve presented in Figure 35, where at least three clear maxima and minima in the potential gravitational energy are observed (blue line in the plot). The case with a relatively moderate elastic intensity of $\epsilon_{CS} = 1 \times 10^{-3}$, also shows a liquid-type of collapse, but it clearly behaves with some solid-like features, observable in the lack of multiple maxima in the energy plot presented in Figure 36.

A simple example of the collapse of a cylindrical column is presented in Figure 37. In this case, the initial particles' lattice has been set in such a way that an intentional slightly unbalanced initial spatial distribution is achieved, allowing the evolution of the collapse to follow a non-symmetrical trajectory. It is important to mention that the simulations with the cylindrical column for the settings used in the prismatic cases offered the same or equivalent type of responses. However, the non-symmetrical evolution captured by this configuration is included here as an example of the flexibility and potential variety of viscoelastic behaviours that can be captured with the methodology proposed in the present work.

Finally, the ability to capture the dynamic behaviour of viscoelastic substances can be better appreciated through animations of the simulations performed using our proposed methodology. Precisely, to get a better grasp of the capabilities of our proposed modelling approach, movies of some of the cases presented in this paper have been produced and made available as supplementary materials. In these videos is possible to appreciate the effect of some of the parameters discussed in this work, and the wide range of possible dynamic response that it is possible to mimic by simply modulating the viscous or elastic components of our model. In Video movie01.mp4 is presented the full evolution represented in Figure 31a,b, whereas in Video movie02.mp4 it is possible to fully appreciate the case presented in Figure 31c,d. The substance column collapse presented in Figure 33a–c can be visualized in Video movie03.mp4, while Videos movie04.mp4 and movie05.mp4 are the animations corresponding to the Figure 33d–i, respectively. The last non-symmetrical column collapse discussed in the previous paragraph, and presented through the snapshots depicted in Figure 37 have been also included in the supplementary material as Video movie06.mp4. From the animations it is clear that our modelling approach is able to represent the wide variety of behaviours and dynamic responses expected from viscoelastic substances.

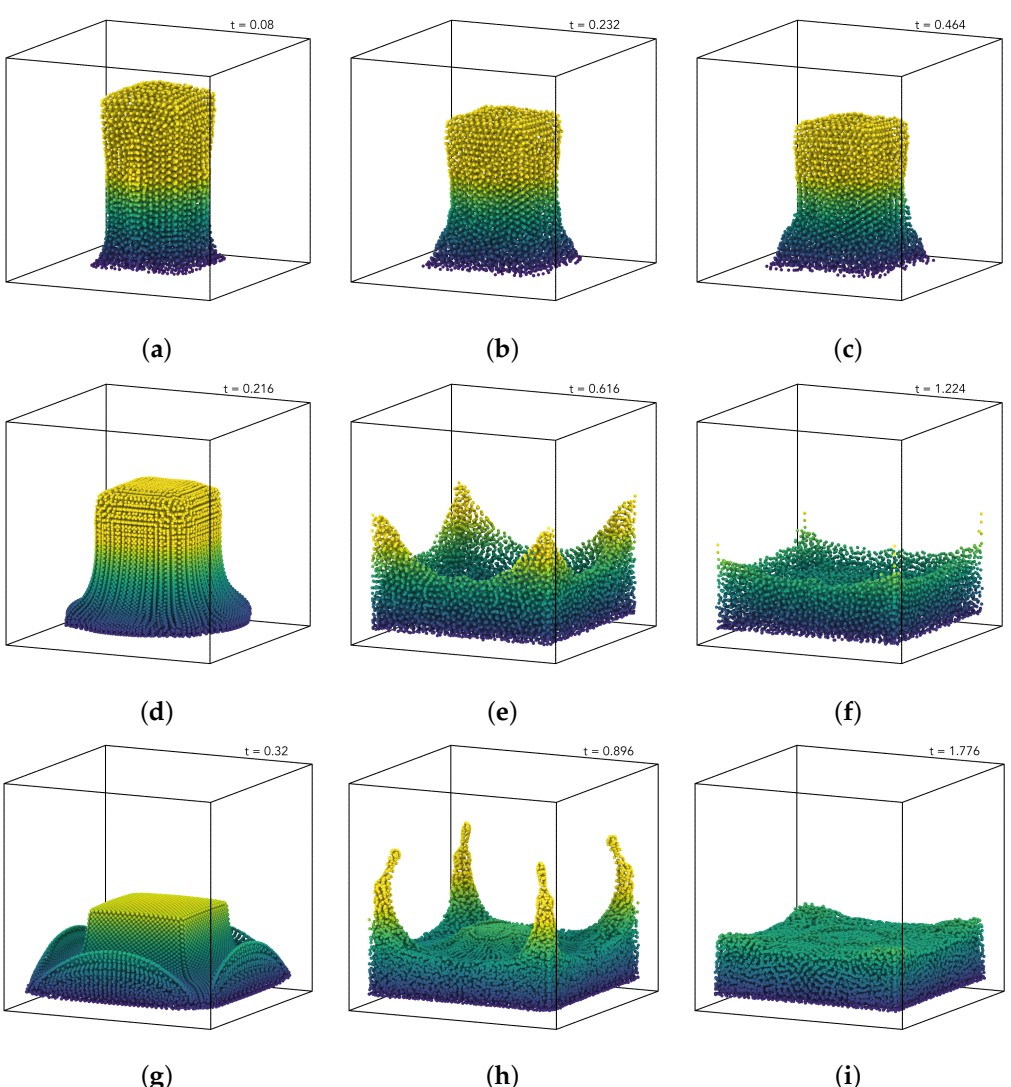

**Figure 33.** Column collapse at three different instants for different ratios of potentials magnitude. (**a**) $\epsilon_{CS} = 1 \times 10^{-2}$, $t \approx 0.20\tau$. (**b**) $\epsilon_{CS} = 1 \times 10^{-2}$, $t \approx 0.55\tau$. (**c**) $\epsilon_{CS} = 1 \times 10^{-2}$, $t \approx 1.1\tau$. (**d**) $\epsilon_{CS} = 1 \times 10^{-3}$, $t \approx 0.20\tau$. (**e**) $\epsilon_{CS} = 1 \times 10^{-3}$, $t \approx 0.55\tau$. (**f**) $\epsilon_{CS} = 1 \times 10^{-3}$, $t \approx 1.1\tau$. (**g**) $\epsilon_{CS} = 1 \times 10^{-4}$, $t \approx 0.20\tau$. (**h**) $\epsilon_{CS} = 1 \times 10^{-4}$, $t \approx 0.55\tau$. (**i**) $\epsilon_{CS} = 1 \times 10^{-4}$, $t \approx 1.1\tau$.

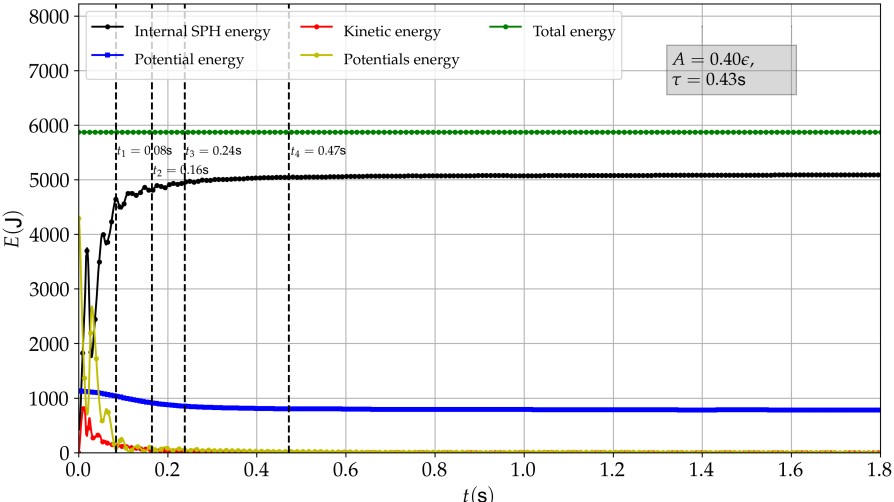

**Figure 34.** Energy budget vs. time for a "mostly" elastic substance. $\epsilon_{CS} = 1 \times 10^{-2}$, $A = 0.4\epsilon_{CS}$.

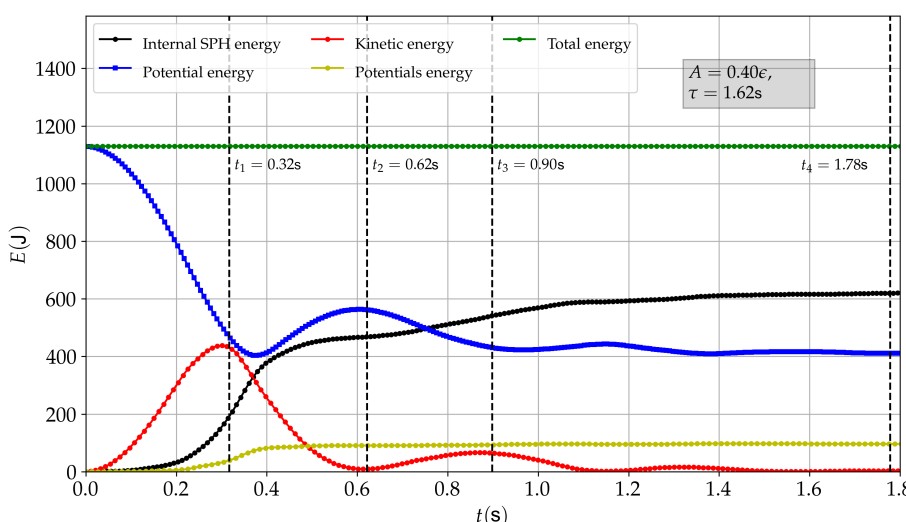

**Figure 35.** Energy budget vs. time for a "mostly" viscous substance. $\epsilon_{CS} = 1 \times 10^{-4}$, $A = 0.4\epsilon_{CS}$.

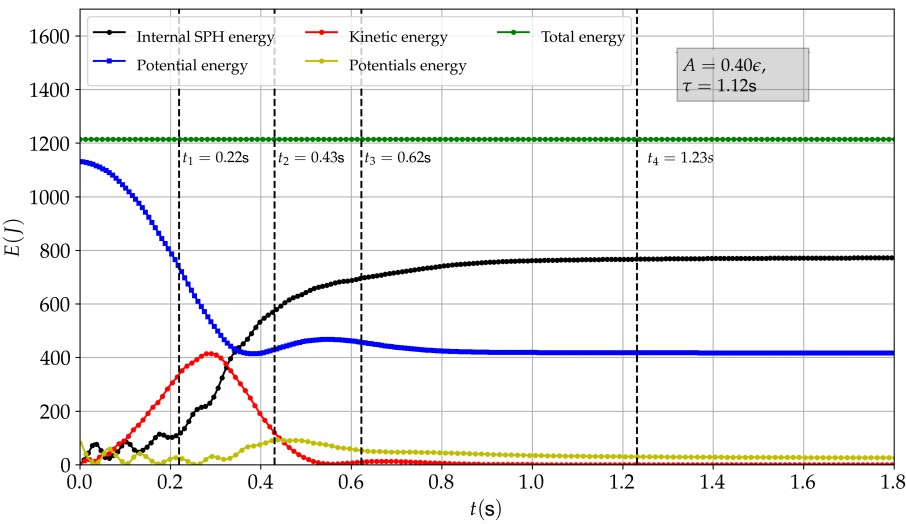

**Figure 36.** Energy budget vs. time for a viscoelastic substance. $\epsilon_{CS} = 1 \times 10^{-3}$, $A = 0.4\epsilon_{CS}$.

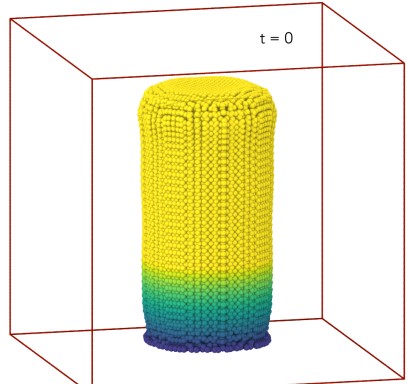
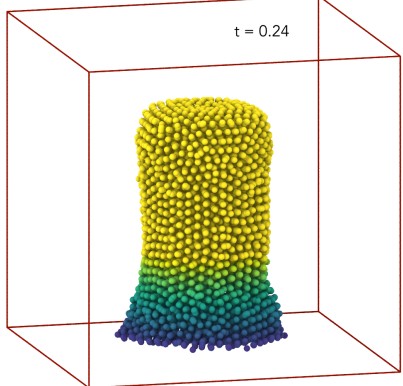

**Figure 37.** *Cont.*

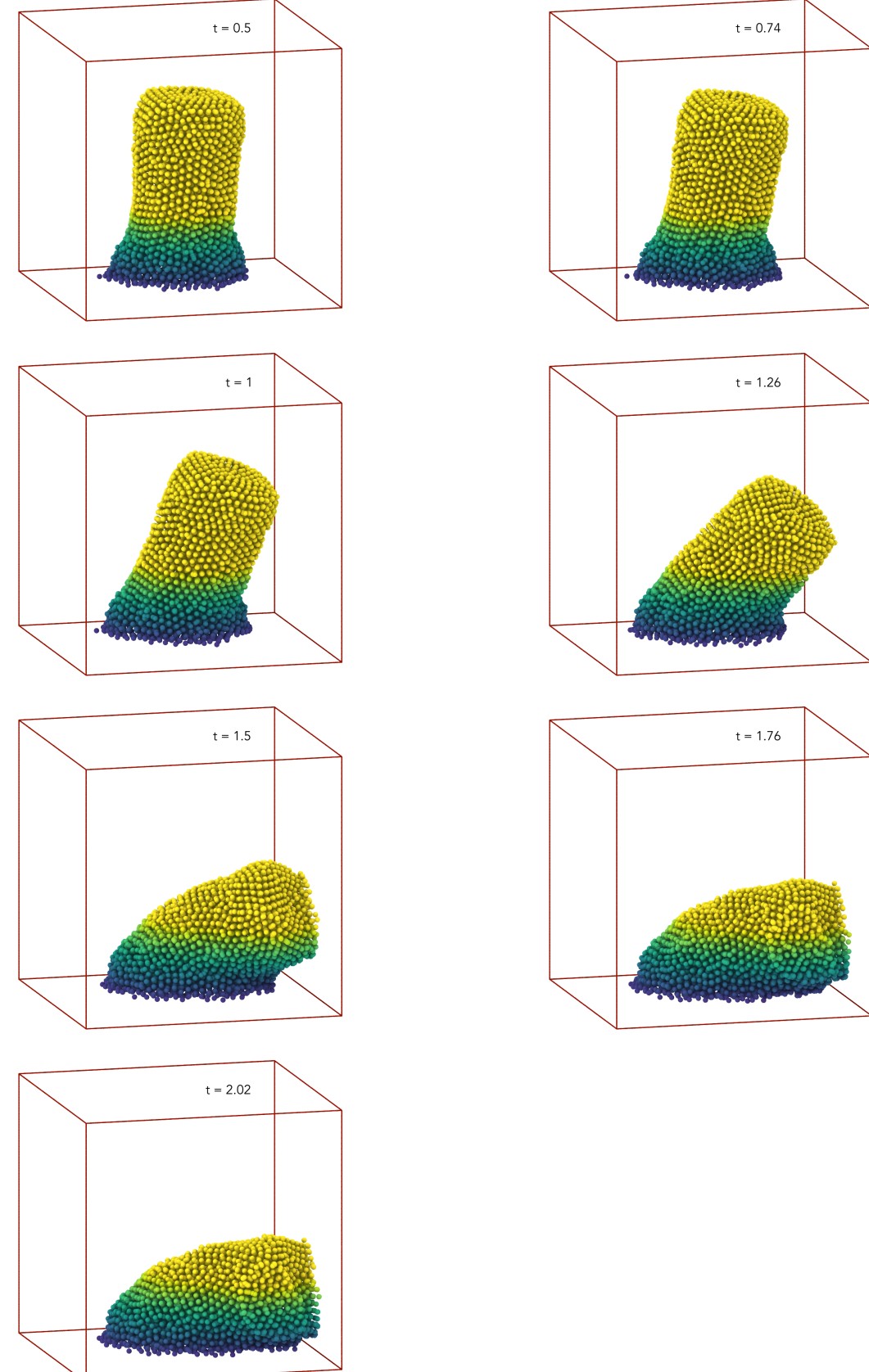

**Figure 37.** Time evolution of a cylindrical column of a viscoelastic substance in a non-symmetrical fashion. Substance settings: $\epsilon_{CS} = 1 \times 10^{-4}$, $\epsilon_{Soft} = 0.2\epsilon_{CS}$, $\rho = 1000$, $c = 10.0$, $\nu_{artf} = 0.8$.

## 6. Conclusions

A simplified methodology for simulating and modelling viscoelastic fluids, based on the concept of linear additive composition of energy potentials, has been proposed, validated, and tested in a wide extent of conditions and applications. The model showed a consistent and clear ability to capture classical features of viscoelastic substances from the rheology perspective. Trends obtained for the Loss and Storage moduli in terms of oscillating frequency and strain showed that the model produces consistent results with some well known and observed characteristics of viscoelastic substances. The methodology and its implementation allowed us to capture with relatively ease the elastic, viscous, and viscoelastic behaviours for different frequencies and strain amplitudes. Some instabilities mostly associated with the equivalent or quasi-elastic potential were identified, but it is clear that an early identification of troublesome combinations of parameters is possible, which would facilitate to any modeller or researcher to search for a more convenient or stable setting.

The model was also tested in a more conventional engineering application, i.e., as a model for flow in a circular pipe of a given viscoelastic substance. In this case, again the model was able to obtain and capture the main features, from the classical parabolic profiles of a pure viscous fluid, to the extreme of a substance showing yield stress, and therefore exhibiting plug flow regime. The model proved to be flexible enough to capture both situations, as well as a number of conditions in between.

The simplified modelling approach was also successful in mimicking the behaviour of substances that might be considered as gels, or extremely viscoelastic, as well as the natural free surface evolution of a liquid column collapsing under the effect of gravity. The simplicity and modularity of the modelling framework proposed in this work suggests that it might be used both as a rigorous simulation tool to study phenomena from a rheological/engineering perspective, as well as a practical modelling tool to emulate the flow of substances that exhibit some level of elasticity. Thanks to the modularity, tunable characteristics of the parameters involved, and conceptual simplicity, the proposed modelling approach can be a powerful simulation tool to be used for researchers and visual graphics modellers alike.

This simplified approach is proposed as a sort of "quick and dirty" method for particle simulations involving viscoelastic materials. If the simulation specifically focuses on the mechanical property the viscoelastic material, we suggest a more rigorous approach (see [53]) that requires rewriting the equation of motion to account for the specific viscoelasticity model. However, if the simulation focuses on the effect of the viscoelastic material on a larger computational domain, the proposed method is easier to implement because it only requires combining together different particle potentials, which is a standard procedure in particle simulations. For instance, in [62], we modelled the watery periciliary layer (PCL) located between the respiratory epithelium and a mucus layer. The PCL is a Newtonian fluid, but mucus has a complex viscoelastic response. In this case, it was important to account for the effect of the mucus layer on the PCL, but an easily implementable approximation of the mucus rheology based on the method proposed here would have been sufficient for the scope of that study.

As a future work, amongst some other possibilities, it would be extremely valuable to perform a clear categorization of relations between geometrical lattice properties, and magnitudes of the potentials used to replicate the quasi-elastic interaction. Furthermore, it is necessary to test the model in more intensive applications, to examine performance and computational costs demanded by any implementation. These factors clearly might help to decide on the more extended use of the technique herein proposed.

**Supplementary Materials:** The following are available online at: https://www.mdpi.com/article/10.3390/chemengineering5030061/s1, Video movie01.mp4: Animation of column collapse simulation using proposed methodology, for a fluid-like substance modelled with $\pi_2 = 9.9 \times 10^2$, $\epsilon_{\text{Soft}} = 0.2\epsilon_{\text{CS}}$; Video movie02.mp4: Animation of column collapse simulation using proposed methodology, for

a fluid-like substance modelled with $\pi_2 = 9.9 \times 10^2$, $\epsilon_{\text{Soft}} = 0.6\epsilon_{\text{CS}}$; Video movie03.mp4: Animation of column collapse for a viscoelastic substance modelled with $\epsilon_{\text{CS}} = 1 \times 10^{-2}$; Video movie04.mp4: Animation of column collapse for a viscoelastic substance modelled with $\epsilon_{\text{CS}} = 1 \times 10^{-3}$; Video movie05.mp4: Animation of column collapse for a viscoelastic substance modelled with $\epsilon_{\text{CS}} = 1 \times 10^{-4}$; Video movie06.mp4: Animation of a cylindrical column collapse of a viscoelastic substance modelled with $\epsilon_{\text{CS}} = 1 \times 10^{-4}$, $\epsilon_{\text{Soft}} = 0.2\,\epsilon_{\text{CS}}$, $\rho = 1 \times 10^3$, c = 10, $\nu_{\text{artf}} = 8 \times 10^{-1}$;

**Author Contributions:** Conceptualization, C.D.-D. and A.A.; methodology, C.D.-D. and A.A.; software, C.D.-D.; validation, C.D.-D.; writing, original draft preparation, C.D.-D.; writing, review and editing, C.D.-D. and A.A.; funding acquisition, C.D.-D. and A.A. All authors read and agreed to the published version of the manuscript.

**Funding:** This research was funded European Union's Horizon 2020 research and innovation programme under the Marie Sklodowska-Curie grant agreement No 841814.

**Institutional Review Board Statement:** Not applicable.

**Informed Consent Statement:** Not applicable.

**Data Availability Statement:** Metadata for all research data created throughout this project has been recorded in the University of Birmingham's current research information system PURE. These records can be searched within the University's Research Portal, FindtIt@Bham.

**Acknowledgments:** The computations described in this paper were performed using the University of Birmingham's BlueBEAR HPC service, which provides a High Performance Computing service to the University's research community. See http://www.birmingham.ac.uk/bear (accessed on 9 August 2021) for more details.

**Conflicts of Interest:** The authors declare no conflict of interest.

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
