# Peer review of "A Simplified Framework for Modelling Viscoelastic Fluids in Discrete Multiphysics"

_2305-7084, doi:10.3390/chemengineering5030061_

Round 1

Reviewer 1 Report

This article aims to model viscoelastic behaviour of fluids by means of a simulative approach. I think the authors did a lot of simulative work, there are a lot of graphic representations and a long discussion, anyway, the results of the simulation are not supported with literature references, neither compared with other modeled data or with experimentally determined ones. The focus of the article it is not clear, how and in which cases these simulations can be useful? Moreover, there are some other lacks.

  • The style and the layout of the paper do not follow the journal template at all.

  • In section 2 “Viscoelastic behaviour and standard methods”, authors mention different models (Maxwell, Kelvin, K-V, Burgers). They should add some references about the application of these models in viscoelastic behaviour modeling.

  • In figure 11, the authors say that looking at the figure it can be observed that (lines 325-326) “the model performs extremely well in a defined range of shear strain amplitude, providing a response that exhibits a clear elastic behaviour”. I do not agree with this statement, data do not show a clear trend as a function of frequency, and the elastic component does not look prevalent on the viscous one. You also do not mention at all which criteria did you follow to choose the used values of ε. A comparison with experimental rheological data is fundamental in this case.

  • Section 5.1, no information about particles, dimension, shape, etc… What kind of fluid are you trying to simulate? For which application? How can this work be useful to interpret and predict the behavior of these fluids? In my opinion, all these important information are not clearly presented and discussed in the paper.

  • The article is too long-winded, the authors discussed a lot about their model and the hypothesis behind it but without clearly present the model itself and the results. I think some discussion can be moved to the introduction section concentrating the following sections on presenting the model and the results.

  • Have you tried to compare the results obtained with experimental ones? Or maybe with data simulated and modeled by means of other models to compare them? Without doing these how can you conclude that citing the authors: “the proposed modelling approach can be a powerful simulation tool for modelling or mimicking the behaviour of viscoelastic substances”?

Attached are also some comments on the text.

Considering my observations I don't think the article deserves to be published. The authors should consider to completely review the work following the suggestions.

Reviewer 2 Report

The authors   method for particle simulations involving  viscoelastic materials focuses on the effect of the viscoelastic material on a large computational domain, which with the proposed method is easier to implement. The paper is well written and includes many previous works.  With this in mind want to bring the followinhg to the authors.

To explore complex visco-elastic materials and to complement any experimental approach new modelling has been proposed and I refer the authors to the spectral approach within different multiphysics framework

Rate type constitutive equations for fiber reinforced nonlinearly vicoelastic solids using spectral invariants

M Shariff, R Bustamante, J Merodio Mechanics Research Communications 84, 60-64, 2017   and     Constitutive modeling framework for residually stressed viscoelastic solids at finite strains NK Jha, J Reinoso, H Dehghani, J Merodio Mechanics Research Communications 95, 79-84, 2019   which complete the review of the bibliography. It is out of the scope of the paper to develop these appproches but should be at least included for completion.

Author Response

We thank and appreciate the kind and valuable comments of the reviewer, as they have allowed us to further improve our manuscript. Please, find our reply and observations to each comment or annotation made on the original paper.

Point 1. To explore complex visco-elastic materials and to complement any experimental approach new modelling has been proposed and I refer the authors to the spectral approach within different multiphysics framework (include paper)

Response 1. We thank the reviewer for bringing into our attention the spectral approach, and its use in modelling viscoelastic solids.  We have included the proposed references in the introduction of our manuscript, together with a very brief discussion about the relevance of the spectral modelling approach.

Reviewer 3 Report

Please see the attached report for comments.

Round 2

Reviewer 1 Report

The authors' adjustments have improved the readability and the quality of the work, which is now acceptable.

Best Regards

Fabio Fanari

Reviewer 3 Report

Authors have satisfactorily addressed my comments and the manuscript in the current form can be accepted for publication.